# Symmetry-Preserving Conformer Ensemble Networks for Molecular Representation Learning

**Yanqiao Zhu**[*]  **Yidan Shi**[*]  **Yuanzhou Chen**  **Fang Sun**  **Yizhou Sun**  **Wei Wang**
Department of Computer Science, University of California, Los Angeles
{yzhu,adrianchen,fts,yzsun,weiwang}@cs.ucla.edu   yidanshi@ucla.edu

## Abstract

Molecular representation learning has emerged as a promising approach for modeling molecules with deep learning in chemistry and beyond. While 3D geometric models effectively capture molecular structure, they typically process single static conformers, overlooking the inherent flexibility and dynamics of molecules. In reality, many molecular properties depend on distributions of thermodynamically accessible conformations rather than single structures. Recent works show that learning from conformer ensembles can improve molecular representations, but existing approaches either produce unphysical structures through averaging or require restrictive molecular alignment. In this paper, we propose Symmetry-Preserving Conformer Ensemble networks (SPiCE), which introduces two key innovations: (1) geometric mixture-of-experts for selective processing of scalar and vector features, and (2) hierarchical ensemble encoding that combines ensemble-level representation with cross-conformer integration. Crucially, SPiCE ensures physically meaningful representations by maintaining joint equivariance to geometric transformations of individual conformers and conformer permutations. Extensive experiments demonstrate that SPiCE consistently outperforms existing conformer ensemble methods and state-of-the-art structural aggregation models across quantum mechanical and biological property prediction tasks.

## 1   Introduction

Molecular Representation Learning (MRL) has emerged as a powerful tool at the intersection of chemistry and machine learning, offering a data-driven approach to encode discrete molecular structures into continuous feature representations [1, 2]. This enables efficient predictions of molecular properties without expensive quantum mechanical calculations for various downstream tasks including drug discovery, materials design, and chemical reaction prediction [3–5].

The field has evolved from early appraoches using simplified molecular representations such as SMILES strings [6] and molecular fingerprints [7, 8] to sophisticated geometric deep learning methods that directly incorporate 2D topological and 3D geometric information [9]. Among these advances, 3D Graph Neural Networks (GNNs) have gained popularity by learning from molecular geometries while respecting fundamental physical symmetries. These models can be categorized as invariant (producing identical outputs regardless of molecular orientation and translation) [10–12], SE(3)–equivariant (transforming consistently under rotations and translations) [13, 14], and E(3)–equivariant (additionally respecting reflection symmetry) [15, 16]. By incorporating these inductive biases, geometric models achieve both improved sample efficiency and better generalization.

However, current 3D MRL models face a fundamental limitation: they typically encode individual conformer structures as if molecules were rigid, static entities. In reality, molecules exist as dynamic

---

[*]These authors made equal contribution to this research.

39th Conference on Neural Information Processing Systems (NeurIPS 2025).

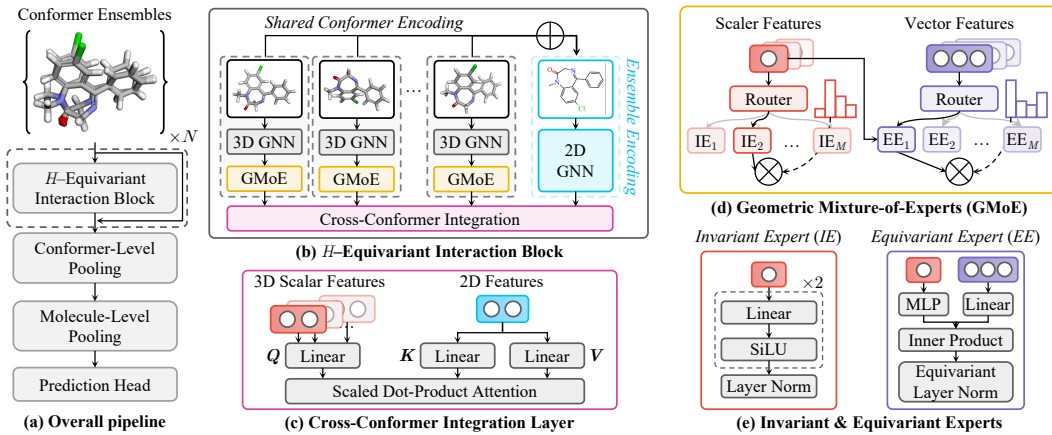

Figure 1: (a) SPiCE processes conformer ensembles through multiple interaction blocks, followed by node-level pooling to obtain conformer representations and ensemble-level pooling for final molecular embeddings. (b) $H$-equivariant interaction block that maintains joint equivariance to conformer permutation ($S_n$) and geometric transformations ($G^n$), composed of shared conformer encoding, geometric mixture-of-experts, and cross-conformer integration. (c) Cross-conformer integration layer combining ensemble encoding through a 2D GNN with attention-based integration. (d) Geometric mixture-of-experts (GMoE) with separate routing for scalar (type-0) and vector (type-1) features. (e) Invariant and equivariant expert networks: equivariant experts preserve rotational symmetry through specialized linear transformations with scalar feature gating for enhanced expressivity.

systems that continuously interconvert between different conformational states through bond rotations, vibrational motions, and intermolecular interactions [17]. Therefore, a conformer ensemble, which refers to the collection of thermodynamically accessible molecular conformations at equilibrium, provides a more complete molecular representation [18]. This is particularly important because many experimentally observable properties depend on the entire distribution of conformers rather than a single static structure. For example, protein-ligand binding often involves conformational selection where proteins recognize specific ligand conformations, and reaction mechanisms frequently depend on the accessibility of particular geometric configurations [19, 20].

Existing approaches to conformer ensemble modeling fall into three categories. Traditional cheminformatics methods like 4D-QSAR [21–23] extend classical mesh-based 3D-QSAR by incorporating conformational information into grid-based molecular descriptors. These methods map molecular properties onto regular 3D lattices for each conformer to produce molecular shape spectra, but require rigid molecular alignment and are limited to datasets with common substructures. Multi-instance learning methods have been adapted from computer vision to treat conformer ensembles as bags of instances [24–28]. However, these approaches typically process conformers independently and then aggregate conformer representations through simple pooling operations, which discard cross-conformer interactions and cannot maintain geometric symmetrices. Additionally, structural aggregation methods [29–31] attempt to combine information from multiple conformers through averaging or clustering procedures. While these can leverage geometric models, they often produce unphysical structures and are highly sensitive to alignment methods.

To address these limitations, we present Symmetry-Preservng Conformer Ensemble networks (SPiCE), a novel model that achieves joint equivariance to both permutations of conformer ensemble ($S_n$) and geometric transformations of individual conformers ($G^n$). SPiCE processes conformer ensembles through a series of $H$-equivariant interaction blocks, where $H = S_n \times G^n$. Each block is composed of three key components: (1) shared conformer encoding with weight-tied 3D GNNs that process each conformer while preserving geometric equivariance, (2) a Geometric Mixture-of-Experts (GMoE) layer that separately routes scalar and vector features through specialized expert networks, enabling type-aware selective processing, and (3) hierarchical ensemble encoding that combines molecular-level context with selective cross-conformer integration through attention mechanisms. Importantly, SPiCE is able to capture both geometric and topological relationships while respecting fundamental geometric symmetries, effectively processing conformer ensembles without requiring alignment or generating unphysical intermediate structures.

We validate SPiCE through comprehensive experiments across diverse molecular property prediction tasks, ranging from quantum mechanical properties to biological activities. Using various 3D GNN backbones and datasets of different scales, we demonstrate consistent improvements over existing conformer ensemble methods. Our analysis reveals scaling behaviors with dataset size, suggesting reliable deployment across different data regimes in real-world applications. Through extensive ablation studies, we justify key architectural choices including the GMoE design, sparse upcycling strategy, and optimal expert granularity for conformer ensemble modeling.

## 2 Preliminaries

### 2.1 Problem Definition

A molecule is represented as a molecular graph $\mathcal{G} = (\mathcal{V}, \mathcal{E}, \boldsymbol{X})$, where $\mathcal{V} = \{v_i\}_{i=1}^{|\mathcal{V}|}$ denotes the node set of atoms, and $\mathcal{E} \subseteq \mathcal{V} \times \mathcal{V}$ represents chemical bonds between atoms. The node attributes are represented in $\boldsymbol{X} \in \mathbb{R}^{|\mathcal{V}| \times d_v}$. For a given molecule, we consider a set of $n$ discrete conformers $\mathcal{C} = \{\boldsymbol{C}_i\}_{i=1}^n$, where each $\boldsymbol{C}_i \in \mathbb{R}^{|\mathcal{V}| \times 3}$ represents atomic 3D coordinates. These conformers are sampled from the thermodynamically-accessible conformational space. Each conformer is associated with a Boltzmann weight $p_i = \exp(-e_i/k_B T) \big/ \sum_j \exp(-e_j/k_B T)$, where $e_i$ is the energy of conformer $\boldsymbol{C}_i$, $k_B$ is the Boltzmann constant, and $T$ is temperature. We note that these Boltzmann weights are *not* provided to the model but rather are used to compute ground-truth ensemble properties.

Our goal is to learn a mapping from the conformer ensemble $(\mathcal{G}, \mathcal{C})$ to molecular properties while preserving both *permutation* invariance of the ensemble and *geometric* symmetries of individual conformers. This requires learning functions that are equivariant to the product group $H = S_n \times G^n$, where $S_n$ is the permutation group over $n$ conformers and $G$ is the geometric symmetry group. Typically, $G$ is taken as the Euclidean group $\mathrm{E}(3)$ of translations, rotations and reflections, or the special Euclidean group $\mathrm{SE}(3)$ of translations and rotations. We allow individual transformations on each conformer individually and independently, hence the overall geometric transformation $G^n$.

### 2.2 Basics of Symmetry Properties

**On permutation group $S_n$.** Our work builds on the characterization of linear $S_n$–equivariant layers [32], which can be decomposed into intra-conformer interactions and a global $G$–invariant aggregation. We formalize this as a theorem and prove it in Appendix C.1. This provides a practical construction of $H$–equivariant layers, by processing individual conformers through $G$–equivariant functions and capturing ensemble interactions via a $G$–invariant function applied to aggregated features. This generalizes Deep Sets [33] to geometric symmetries in molecular conformers.

**On geometry group $G$.** We decompose geometric transformations into translations and rotations (including reflections for $E(3)$): $G = T \rtimes R$. Following Equiformer [34], we use *type-0* and *type-1* to express $R$–invariant and equivariant features and list basic symmetry-preserving operations below.

**Lemma 1.** *The following operations preserve symmetry:*

*(1) Any function $f(s)$ of an $R$–invariant feature $s$ is $R$–invariant;*
*(2) The product $s \cdot \boldsymbol{v}$ between $R$–invariant feature $s$ and $R$–equivariant feature $\boldsymbol{v}$ is $R$–equivariant.*
*(3) The inner product $\langle \boldsymbol{v}_1, \boldsymbol{v}_2 \rangle$ between two $R$–equivariant features $\boldsymbol{v}_1$, $\boldsymbol{v}_2$ is $R$–invariant.*

## 3 Method

The overall architecture of SPiCE is illustrated in Figure 1. SPiCE employs a hierarchical architecture that enables structural interaction across atoms and conformers. Its core is a series of $H$–equivariant interaction blocks designed to process conformer ensembles while maintaining the relevant symmetries. Intra-conformer learning is performed using weight-shared equivariant 3D GNNs, which jointly handle scalar and vector features. To support type-specific and selective information processing, we design a Geometric Mixture-of-Experts (GMoE) module. The resulting features are then pooled to form an ensemble representation with 2D GNNs that capture the topological structure of the molecular system. These molecule- and conformer-level features are subsequently combined through a cross-conformer integration module. Following the interaction blocks, we apply node-level pooling within each conformer to generate conformer-level embeddings and conformer-level pooling

across the ensemble to obtain the final molecular embedding. This embedding is then passed through a prediction head to produce the output property prediction. A PyTorch-like pseudocode of SPiCE is provided in Appendix B.

**Overall $H$–equivariance.** We specify the input and output invariance and equivariance types for every module in the following sections. Since each module preserves symmetry and the output symmetrical properties of each module match the input properties of each subsequent module, the overall symmetry with respect to $H$ is guaranteed through the composition of all modules.

**Other remarks.** We batch over both conformers and molecules, hence our features shapes often start with $\mathbb{R}^{n \times |\mathcal{V}| \times \cdots}$. We denote $N = n \times |\mathcal{V}|$ as the total number of atoms across all conformers and interchangeably use $\mathbb{R}^{N \times \cdots}$ to express dimensionality. Typically, the second-to-last dimension of our features (dim=-2) is either 1 for $R$–invariant scalar features or 3 for $R$–equivariant vector features. We often concatenate these features into a single **hybrid** feature with size 4 over this dimension, and call a hybrid feature $R$–equivariant when the corresponding type-0 feature is $R$–invariant and type-1 feature is $R$–equivariant. Many modules of our model perform on each conformer *separately*, in which case $S_n$–equivariance automatically holds, and individual $G$–symmetries lead to global $G^n$–symmetry.

## 3.1 Input Representation and Feature Construction

For a conformer ensemble, each $H$–equivariant interaction block operates on graph node features that combine atomic and geometric information. For a molecule with $m$ atoms, the input consists of: (a) atomic numbers $z \in \mathbb{Z}_+^{|\mathcal{V}|}$, which are embedded into initial node features $X \in \mathbb{R}^{|\mathcal{V}| \times d}$ through a learnable embedding layer, and (b) atomic coordinates $\{c_i\}_{i=1}^m$ where $c_i \in \mathbb{R}^3$. From the coordinates, we compute relative position vectors $\rho_{ij} = c_i - c_j$.

The $H$–equivariant interaction blocks process both features $x_i$ and geometric vectors $\rho_{ij}$ to construct messages that respect the underlying symmetries. To encode distance information effectively, we transform the relative distances $r = |\rho_{ij}|$ using a set of radial basis functions. Specifically, we employ Gaussian radial basis functions (RBF) $\phi_k(r) = \exp(-\gamma(r - \mu_k)^2)$, where $\{\mu_k\}_{k=1}^K$ are equally spaced centers and $\gamma$ controls the width of the Gaussians [10]. These radial features are then processed through a three-layer neural network, with layer normalization [35] and SiLU activation [36] after the first two linear layers. The output of this network parametrizes the message construction in the interaction blocks. Thanks to the RBF positional encoding, our encoded features are all $T$–**invariant**, therefore the $T^n$–invariance of the entire model is already guaranteed.

## 3.2 $H$–Equivariant Interaction Blocks

As the model is equipped with $L$ interaction blocks, here we analyze the architecture within the $l$-th interaction block, and mostly omit the layer index $l$ in the following discussions for simplicity.

### 3.2.1 Shared Conformer Encoding

With a conformer set with hybrid node feature as $\mathbf{X} = (\mathbf{X}^s, \mathbf{X}^v) \in \mathbb{R}^{n \times |\mathcal{V}| \times 4 \times d}$, we employ shared 3D GNN layers to process conformer geometric information. A 3D GNN layer consists of two key learnable functions: message construction $\Phi$ and feature update $\Psi$, which can be formulated as

$$h_i = \Psi\left(x_i, \bigoplus_{j \in \mathcal{N}_i} \Phi(x_i, x_j, c_{ij})\right), \qquad (1)$$

where $h_i = (h_i^s, h_i^v)$, $h_i^s \in \mathbb{R}^{1 \times d}$ represents the scaler (type-0, rotationally invariant) feature for atom $i$ with 1 tensor component and $d$ channels, and $h_i^v \in \mathbb{R}^{3 \times d}$ represents the vector (type-1, rotationally equivariant) feature for atom $i$ with 3 tensor components and $d$ channels. For atom $i$ in an arbitrary conformer, $\mathcal{N}_i$ denotes the set of its neighbors, $x_i = (x_i^s, x_i^v) \in \mathbb{R}^{4 \times d}$ represent the concatenated scalar and vector features, $c_{ij} = (c_{ij}^s, c_{ij}^v) \in \mathbb{R}^{4 \times d_m}$ are concatenated scalar and vector messages between atom $i$ and $j$, $d, d_m$ are feature dimensions, and $\bigoplus$ denotes message aggregation over neighbors $\mathcal{N}_i$ (typically summation).

For the first interaction block ($l = 0$), $\mathbf{X}^{\mathrm{s}}$ (type-0) contains initial molecular features including embeddings of atomic numbers $\mathbf{z}$, distance-based radial basis functions, and topological descriptors, while $\mathbf{X}^{\mathrm{v}}$ (type-1) contains initial geometric features including embeddings of relative position vectors and directional bond features. For subsequent interaction blocks ($l > 0$), the features include both the initial features above plus the processed outputs $\widetilde{\mathbf{H}}^{\mathrm{s}}$ and $\widetilde{\mathbf{H}}^{\mathrm{v}}$ from the previous interaction block.

The 3D GNN interaction layer outputs hybrid features $\mathbf{H} = (\mathbf{H}^{\mathrm{s}}, \mathbf{H}^{\mathrm{v}}) \in \mathbb{R}^{n \times |\mathcal{V}| \times 4 \times d}$, where $\boldsymbol{h}_i$ denotes the hybrid node embedding for the $i$-th atomic node within this tensor. We also process $\mathbf{H}$ with a residual connection and an Equivariant Layer Normalization (ELN), which is a generalized layer normalization for $G$–equivariant features [34]. Here each conformer is operated separately, therefore the output $\mathbf{H}$ is $S_n$–**equivariant**; plus, all 3D GNNs are pre-selected to be equivariant, so it is also independently $R$–**equivariant** for each conformer.

We note that our framework is agnostic to the specific architecture of the message and update functions. They can be either Cartesian equivariant models (e.g., PaiNN [37]) that operate directly in Cartesian coordinates to construct messages and update features, or spherical equivariant models that employ spherical harmonics and tensor products for message passing (e.g., Equiformer [34]), with channel-mixing and gated non-linear functions for feature updates.

### 3.2.2 Geometric Mixture-of-Experts (GMoE)

When processing molecular conformers, certain atoms or structural features may be more relevant or important than others for specific properties. Graph pooling approaches [38–40] achieve this selective processing by focusing computation on important nodes through attention mechanisms. This naturally extends to Mixture-of-Experts (MoE) architectures [41, 42], which specialize multiple neural networks for selective information passing with a learnable routing mechanism.

Since scalar and vector features require distinct handling due to their different transformation properties under symmetry operations, we propose GMoE as shown in Figure 1(d) that maintains equivariance for vector features while ensuring invariance for scalar features through distinct router and expert groups. Note that all operations in this section operate on each node (and hence each conformer) separately, therefore $S_n$–**equivariance** is already guaranteed, and any claim on rotational $R$– invariance / equivariance automatically extends to $R^n$.

GMoE processes the two types of features in $\mathbf{H} = (\mathbf{H}^{\mathrm{s}}, \mathbf{H}^{\mathrm{v}}) \in \mathbb{R}^{N \times 4 \times d}$ using separate modules. For a given node $i$, let $\boldsymbol{h}_i^{\mathrm{s}} \in \mathbb{R}^{1 \times d}$ and $\boldsymbol{h}_i^{\mathrm{v}} \in \mathbb{R}^{3 \times d}$ denote its scalar and vector features, respectively. We extend the standard Mixture of Experts (MoE) formulation [43, 44] to handle scalar and vector features independently, introducing two sets of routers and experts that process invariant and equivariant features respectively. The numbers of invariant and equivariant experts are denoted by $N_{\mathrm{I}}$ and $N_{\mathrm{E}}$.

**Permutation-equivariant router design.** For each node $i$, the concatenated routing weights $\hat{\boldsymbol{r}}_i = (\hat{\boldsymbol{r}}_i^{\mathrm{s}}, \hat{\boldsymbol{r}}_i^{\mathrm{v}}) \in \mathbb{R}^{N_{\mathrm{I}} + N_{\mathrm{E}}}$ for scalar and vector branches are computed through a two-stage process. First, the corresponding routing networks $r^{\mathrm{s}}(\boldsymbol{h}_i^{\mathrm{s}})$ and $r^{\mathrm{v}}(\boldsymbol{h}_i^{\mathrm{v}})$, both with $R$–**invariant** outputs, compute initial scores $\boldsymbol{r}_i = (\boldsymbol{r}_i^{\mathrm{s}}, \boldsymbol{r}_i^{\mathrm{v}}) \in \mathbb{R}^{N_{\mathrm{I}} + N_{\mathrm{E}}}$ of $(N_{\mathrm{I}}, N_{\mathrm{E}})$ experts for scalar and vector features. Following SAGPool [38], our scalar router leverages a GCN layer [45] to compute routing scores, and our vector router calculates a linearly transformed inner product of features:

$$r^{\mathrm{s}}(\boldsymbol{h}_i^{\mathrm{s}}) = \mathrm{softmax}\left(\sum_{j \in \mathcal{N}(i)} \boldsymbol{h}_j^{\mathrm{s}} \boldsymbol{W}^{\mathrm{s}} / \sqrt{d_i d_j} + \boldsymbol{b}\right), \tag{2}$$

$$r^{\mathrm{v}}(\boldsymbol{h}_i^{\mathrm{v}}) = \mathrm{softmax}\left(\boldsymbol{h}_i^{\mathrm{v}\top} \boldsymbol{h}_i^{\mathrm{v}} \boldsymbol{W}^{\mathrm{v}}\right). \tag{3}$$

where $d_i$ represents the number of neighbors of node $i$. The $R$-**invariance** of $r^{\mathrm{v}}$ is due to (3) of Lemma 1: operation $\boldsymbol{h}_i^{\mathrm{v}\top} \boldsymbol{h}_i^{\mathrm{v}}$ performs inner products over the equivariant dimension. The scores then undergo top-$k$ selection and normalization:

$$r'^{\mathrm{s}}_{e,i} = \begin{cases} r^{\mathrm{s}}_{e,i}, & \text{if } r^{\mathrm{s}}_{e,i} \in \text{top-}k\left(\{r^{\mathrm{s}}_{e,i}\}_{e=1}^{N_{\mathrm{I}}}\right), \\ 0, & \text{otherwise}, \end{cases} \qquad \hat{r}^{\mathrm{s}}_{e,i} = \frac{r'^{\mathrm{s}}_{e,i}}{\sum_{e=1}^{k} r'^{\mathrm{s}}_{e,i}}, \tag{4}$$

where $r^{\mathrm{s}}_{e,i}$ refers to the score for the $i$-th atom of the $e$-th expert, and $k$ is the pre-set number of selected experts for each type of hybrid features. The computation for $\hat{r}^{\mathrm{v}}_{e,i}$ shares the same logic as $\hat{r}^{\mathrm{s}}_{e,i}$. The resulting $\{\hat{\boldsymbol{r}}_i = (\hat{\boldsymbol{r}}_i^{\mathrm{s}}, \hat{\boldsymbol{r}}_i^{\mathrm{v}})\}_{i \in \mathcal{C} \times \mathcal{V}}$ is $R^n$–**invariant** from (1) of Lemma 1.

**Invariant and equivariant experts.** Each Invariant Expert (IE) is implemented as a two-layer MLP with SiLU activations followed by layer normalization. The output of the $e$-th IE can be defined as $\mathbf{F}_e^s = f_e^s(\mathbf{H}^s) \in \mathbb{R}^{N \times 1 \times d}$. These outputs are also $R^n$–**invariant** from (1) of Lemma 1. Meanwhile, for maintaining geometric symmetries, each Equivariant Expert (EE) is formulated as:

$$\boldsymbol{f}_{e,i}^v = \boldsymbol{h}_i^v \boldsymbol{W}_e \, \text{diag}(\text{MLP}(\boldsymbol{h}_i^s)), \tag{5}$$

where $\boldsymbol{f}_{e,i}^v$ is the output of the $e$-th expert on node $i$, and the concatenated output $\mathbf{F}_e^v = f_e^v(\mathbf{H}^v, \mathbf{H}^s) \in \mathbb{R}^{N \times 4 \times d}$. Here $\boldsymbol{W}_e \in \mathbb{R}^{d \times d}$ is a learnable weighting matrix, and $\text{diag}$ constructs a diagonal matrix from a vector. EE is $R^n$–**equivariant** from (2) of Lemma 1 since $\boldsymbol{h}_i^v$ is equivariant, while both $\boldsymbol{W}_e$ and $\text{diag}(\text{MLP}(\boldsymbol{h}_i^s))$ are invariant, the latter being a consequence of (1) of Lemma 1.

**Node-wise weighted summarization.** With selected expert weights and expert output for both scalar and vector features, a node-wise weighted summarization is employed for dynamic feature ensemble. For node $i \subseteq \mathcal{C} \times \mathcal{V}$, the scalar and vector representations can be calculated as:

$$\widetilde{\boldsymbol{h}}_i^s = \sum_e \hat{r}_{e,i}^s \boldsymbol{f}_{e,i}^s : \mathbb{R}^{1 \times d}, \qquad \widetilde{\boldsymbol{h}}_i^v = \sum_e \hat{r}_{e,i}^v \boldsymbol{f}_{e,i}^v : \mathbb{R}^{3 \times d}. \tag{6}$$

For a molecular conformer ensemble, the output of MoE module is represented as $\widetilde{\mathbf{H}} = (\widetilde{\mathbf{H}}^s, \widetilde{\mathbf{H}}^v) \in \mathbb{R}^{N \times 4 \times d}$, where the $\widetilde{\mathbf{H}}^s$ and $\widetilde{\mathbf{H}}^v$ are $R^n$–invariant and $R^n$–equivariant, respectively from (1) and (2) of Lemma 1. Hence, the hybrid feature $\widetilde{\mathbf{H}}$ is $R^n$–**equivariant**.

**Router training and regularization.** To ensure balanced expert utilization, we employ three key mechanisms: (1) Router z-loss to punish extreme routing decisions [46]:

$$\mathcal{L}_R(\boldsymbol{r}) = \frac{1}{N} \sum_{i=1}^N \left( \log \sum_{e=1}^{N_I} \exp(r_{e,i}^s)^2 + \log \sum_{e=1}^{N_E} \exp(r_{e,i}^v)^2 \right), \tag{7}$$

where $r_{e,i}^v$ and $r_{e,i}^s$ represents the $R^n$–invariant score from invariant and equivariant routers for the $i$-th node of the $e$-th expert, and $N_I$ and $N_E$ are the number of invariant and equivariant experts.

(2) Top-$k$ expert selection with Gumbel-Sigmoid sampling [47, 48], where the Gumbel noise allows the experts to be better differentiated in the process of training:

$$\text{Gumbel-Sigmoid}(x) = \text{Sigmoid}(x + G' - G''), \tag{8}$$

where routing logits $r_{i,e}^s$ or $r_{i,e}^v$ are input $x$, and $G'$ and $G''$ are independent Gumbel noise samples.

(3) Expert upcycling strategy [49] that gradually increases the number of active experts during training. Upcycling combined with the Gumbel-Sigmoid technique allows us to overcome the limitations of static expert selection and achieve improved performance in our geometry-aware sparse architecture.

### 3.2.3 Ensemble Encoding

The ensemble encoding block enables cross-conformer interactions while preserving $S_n$–equivariance. Let $\bar{\mathbf{H}} = (1/n) \sum_{c=1}^n \widetilde{\mathbf{H}}_c^s \in \mathbb{R}^{|\mathcal{V}| \times 1 \times d}$ denote the mean-pooled representation of $S_n$–equivariant, $R^n$–invariant scalar features $\widetilde{\mathbf{H}}^s$ across all conformers, which is $S_n$–**invariant** and $R^n$–**invariant**. This is processed by a Graph Isomorphism Network (GIN) layer [50], incorporated with 2D topological graph edge information. The propagation rule updates node $i$ embeddings as:

$$\bar{\boldsymbol{h}}_i = \varphi \left( (1 + \epsilon)\bar{\boldsymbol{h}}_i + \sum_{j \in \mathcal{N}(i)} \bar{\boldsymbol{h}}_j \right), \tag{9}$$

where $\epsilon$ is learnable and $\varphi$ is a two-layer perceptron with ReLU activation. Let $\bar{\mathbf{H}}^s \in \mathbb{R}^{|\mathcal{V}| \times 1 \times d}$ denote the final node embedding matrix after the GIN interaction layer, which aggregates information across the conformer ensemble, and is $S_n$–**invariant** and $R^n$–**invariant** from (1) of Lemma 1.

### 3.2.4 Gated Aggregation

While the GMoE mechanism handles self-attention within each conformer, we employ gated aggregation to facilitate information passing between conformer-level and molecular-level features.

This cross-attention mechanism integrates *conformer-level* features $\mathbf{H}^{\mathrm{s}} \in \mathbb{R}^{(n \times |\mathcal{V}|) \times 1 \times d}$ and the aggregated *molecular-level* GIN output features $\overline{\mathbf{H}}^{\mathrm{s}} \in \mathbb{R}^{|\mathcal{V}| \times 1 \times d}$. For ease of discussion, we squeeze the scalar dimension for both representations (dim=-2) in the following matrix multiplications. The gated integration is defined as $\mathbf{X}^{\mathrm{s}} = \mathrm{Attn}(\mathbf{H}^{\mathrm{s}}, \overline{\mathbf{H}}^{\mathrm{s}})$, where the attention mechanism computes:

$$\boldsymbol{Q} = \overline{\mathbf{H}}^{\mathrm{s}} \boldsymbol{W}_Q, \qquad \boldsymbol{K} = \mathbf{H}^{\mathrm{s}} \boldsymbol{W}_K, \qquad \boldsymbol{V} = \mathbf{H}^{\mathrm{s}} \boldsymbol{W}_V, \qquad (10)$$

$$\mathrm{Attn}(\mathbf{H}^{\mathrm{s}}, \overline{\mathbf{H}}^{\mathrm{s}}) = \mathrm{softmax}\left(\frac{\boldsymbol{Q}\boldsymbol{K}^{\top}}{\sqrt{d_k}}\right)\boldsymbol{V}. \qquad (11)$$

Here, $\mathbf{W}_Q \in \mathbb{R}^{d \times d_k}, \mathbf{W}_K \in \mathbb{R}^{d \times d_k}, \mathbf{W}_V \in \mathbb{R}^{d \times d}$ are learnable projection matrices, $d_k$ is the dimension of the attention space. After reshaping, we have the output $\mathbf{X}^{\mathrm{s}} \in \mathbb{R}^{n \times |\mathcal{V}| \times 1 \times d}$. This mechanism enables each conformer to selectively incorporate molecular-level and inter-conformer information, facilitating better ensemble representation learning. Since every input here are $R^n$–invariant, the output $\mathbf{X}^{\mathrm{s}}$ is also $R^n$–**invariant** from (1) of Lemma 1. We further prove in Section C.2 that $\mathbf{X}^{\mathrm{s}}$ is also $S_n$–**equivariant**.

Finally, we denote $\mathbf{X}^{(l+1)} = \left[\mathbf{X}^s, \widetilde{\mathbf{H}}^v\right] \in \mathbb{R}^{N \times 4 \times d}$ as the next-level hybrid input feature that satisfies $S_n$–**equivariant** and $R^n$–**equivariant**, thus completing our single $H$–equivariant layer architecture.

### 3.3 Model Training

The proposed architecture consists of an equivariant graph neural network with $L$ interaction layers. Each interaction layer contains $N_{\mathrm{I}}$ invariant experts and $N_{\mathrm{E}}$ equivariant experts to process type-0 and type-1 feature separately. For regression tasks, the objective is a weighted sum of the standard Mean Squared Error (MSE) loss and an auxiliary z-loss term. For classification tasks, a binary classification head equipped with sigmoid activation is attached to the final molecular representations and we use a binary cross entropy loss for model training.

## 4 Experiments

In this section, we present comprehensive experimental evaluations of SPiCE across multiple molecular datasets and tasks. We first describe our datasets and experimental setup, then present results and analyses. We aim to answer the following research questions:

- **RQ1: Performance.** How does SPiCE perform across different molecular property prediction tasks compared to state-of-the-art conformer ensemble methods?
- **RQ2: Scalability.** How does the performance of SPiCE scale with dataset size?
- **RQ3: Architecture design.** What is the impact of key architectural choices in SPiCE?

### 4.1 Experimental Setup

**Datasets.** We evaluate SPiCE on four datasets spanning both regression and classification tasks: (1) Drugs-7.5K, obtained by downsampling 10% of Drugs-75K [51] with a fixed random seed due to computational constraints, with three quantum mechanical properties: IP, EA, and $\chi$, (2) Kraken [52] with four 3D ligand descriptors: (Sterimol $B_5$, Sterimol L, BurB$_5$, BurL), and (3-4) CoV2 and CoV2-3CL from GEOM-Drugs [53]: CoV2 measures general inhibition in human cells, while CoV2-3CL specifically targets the 3CL protease inhibition. Prior to training, we perform preprocessing including conformer deduplication, clustering, and selection following the methodology described in prior works [30, 51]. We present detailed statistics and descriptions of datasets in Appendix E.

**Metrics.** Following prior works, we use Mean Absolute Error (MAE) for regression tasks and Receiver Operating Characteristic (ROC) area under the curve for classification tasks, particularly appropriate for the highly imbalanced datasets CoV2 and CoV2-3CL.

**Baselines.** We evaluate SPiCE against three categories of baselines. (1) 1D string-based and 2D topological models, including fingerprints with random forest [54], extended 3D fingerprints with random forest [55], and 2D GNN models (GIN [50], GIN with virtual nodes [56], and GraphGPS [57]). (2) 3D single-conformer GNNs with random conformer sampling [51], including PaiNN [37], ViSNet [58], ClofNet [13], and Equiformer [34]. (3) Conformer ensemble methods: For each 3D GNN backbone, we implement three conformer aggregation strategies following [51]: mean

Table 1: Performance in terms of MAE (↓) for seven regression tasks (Drugs-7.5K, Kraken) and ROC scores (↑) for two classification tasks (CoV2, CoV2-3CL). **Bold** and underlined values indicate best and second-best overall performance, respectively.

| Backbone | Ensemble Strategy | Drugs-7.5K (MAE, ↓) | | | Kraken (MAE, ↓) | | | | CoV2 ROC (↑) | 3CL ROC (↑) |
|---|---|---|---|---|---|---|---|---|---|---|
| | | IP | EA | $\chi$ | $B_5$ | L | $BurB_5$ | BurL | | |
| ► *1D String-based and 2D Topological Approaches* | | | | | | | | | | |
| Fingerprint+RF [54] | | 0.5833 | 0.5277 | 0.3130 | 0.4760 | 0.4303 | 0.2758 | 0.1521 | 0.6071 | 0.9013 |
| E3FP+RF [55] | | 0.6217 | 0.5774 | 0.3464 | 0.6249 | 0.5535 | 0.3692 | 0.1908 | 0.6046 | 0.7676 |
| GIN [50] | | 0.5575 | 0.5116 | 0.2892 | 0.3128 | 0.4003 | 0.1719 | 0.1200 | 0.3708 | 0.5942 |
| GIN-VN [56] | | 0.5398 | 0.5160 | 0.2937 | 0.3567 | 0.4344 | 0.2422 | 0.1741 | 0.4832 | 0.7387 |
| GraphGPS [57] | | 0.5480 | 0.5054 | 0.2863 | 0.3450 | 0.4363 | 0.2066 | 0.1500 | 0.5601 | 0.8387 |
| ► *3D Single-Conformer Graph Neural Networks with Random Conformer Sampling [51]* | | | | | | | | | | |
| PaiNN [37] | | 0.5557 | 0.5127 | 0.2924 | 0.3443 | 0.4471 | 0.2395 | 0.1673 | 0.2997 | 0.8368 |
| ClofNet [13] | | 0.6316 | 0.6008 | 0.3615 | 0.4473 | 0.6369 | 0.3216 | 0.2426 | 0.5233 | 0.7562 |
| Equiformer [34] | | 0.5471 | 0.4898 | 0.2887 | 0.2709 | 0.3759 | 0.2019 | 0.1526 | 0.4577 | 0.8035 |
| ViSNet [58] | | 0.5393 | 0.4855 | 0.2985 | 0.3828 | 0.4495 | 0.2400 | 0.1755 | 0.5011 | 0.4774 |
| ► *Conformer Ensemble Approaches* | | | | | | | | | | |
| ConfNet [60] | | 0.5760 | 0.5359 | 0.3057 | 0.4469 | 0.4680 | 0.2686 | 0.1657 | 0.5010 | 0.4930 |
| ConAN-FGW [31] | | 0.5471 | 0.4945 | 0.2891 | 0.3242 | 0.5178 | 0.2026 | 0.1492 | 0.6340 | 0.9180 |
| PaiNN [37] | Mean | 0.5410 | 0.4966 | 0.2963 | 0.2877 | 0.3950 | 0.1817 | 0.1472 | 0.5722 | 0.8850 |
| | DeepSets | 0.5396 | 0.5091 | 0.2982 | 0.2225 | 0.3619 | 0.1693 | 0.1324 | 0.5802 | 0.6808 |
| | Attention | 0.6318 | 0.5985 | 0.3488 | 0.3496 | 0.4109 | 0.2123 | 0.1506 | 0.4179 | 0.6984 |
| | SPiCE | 0.5281 | 0.4929 | **0.2792** | 0.2178 | 0.3548 | **0.1564** | 0.1292 | 0.5910 | 0.8880 |
| ClofNet [13] | Mean | 0.5935 | 0.5441 | 0.3121 | 0.3986 | 0.5674 | 0.2857 | 0.2327 | 0.3900 | 0.7580 |
| | DeepSets | 0.5912 | 0.5533 | 0.3153 | 0.3314 | 0.5375 | 0.2532 | 0.1983 | 0.6208 | 0.7628 |
| | Attention | 0.6694 | 0.5949 | 0.3578 | 0.4979 | 0.6118 | 0.3353 | 0.2502 | 0.3707 | 0.8182 |
| | SPiCE | 0.5747 | 0.5283 | 0.3059 | 0.3193 | 0.4903 | 0.2477 | 0.1913 | 0.6730 | **1.0000** |
| Equiformer [34] | Mean | 0.5457 | 0.4932 | 0.2977 | 0.2303 | 0.3830 | 0.1680 | 0.1259 | 0.5601 | 0.8387 |
| | DeepSets | 0.5404 | 0.4888 | 0.2990 | 0.2564 | 0.3772 | 0.1782 | 0.1234 | 0.5125 | 0.7134 |
| | Attention | 0.5488 | 0.4923 | 0.2896 | 0.3187 | 0.4508 | 0.1673 | 0.1425 | 0.3882 | 0.7881 |
| | SPiCE | 0.5318 | 0.4830 | 0.2816 | 0.2241 | **0.3456** | 0.1611 | **0.1229** | 0.5650 | 0.8405 |
| ViSNet [58] | Mean | 0.5593 | 0.4927 | 0.2862 | 0.2811 | 0.3970 | 0.1874 | 0.1469 | 0.6035 | 0.7447 |
| | DeepSets | **0.5280** | 0.4987 | 0.2846 | 0.3104 | 0.4113 | 0.1716 | 0.1314 | 0.6626 | 0.4160 |
| | Attention | 0.5593 | 0.4988 | 0.2944 | 0.3755 | 0.4195 | 0.2384 | 0.1394 | 0.5262 | 0.7158 |
| | SPiCE | 0.5384 | **0.4538** | 0.2814 | **0.2715** | 0.3807 | 0.1657 | 0.1277 | **0.6890** | 0.7195 |

pooling, DeepSets [33], and self-attention [59]. We also compare against specialized ensemble models: ConfNet [60] and ConAN-FGW [31]. Please refer to Appendix F for details of baselines.

Note that SPiCE is designed as a plug-and-play framework that requires compatible backbones with equivariant representations. While we primarily use equivariant models due to their higher representational capabilities, we also evaluate invariant 3D models as baselines in Appendix H.

**Experimental settings.** Following prior works, for regression datasets, we randomly partition data into training, validation, and test sets with a 7:1:2 ratio, while classification datasets use fixed public splits. We optimize models using AdamW [61] with a cosine decay scheduler. Training terminates if the loss shows no improvement for 400 consecutive epochs. To ensure consistent comparison, we set the latent feature dimension to 128 and limit each molecule to a maximum of 20 conformers. For classification tasks, we address label imbalance by maintaining a 1:1 ratio of positive to negative samples during training. Other configurations follow the original settings from respective papers. The details of experimental settings are provided in Appendix D.

## 4.2 Main Results (RQ1)

Table 1 summarizes the performance across all tasks. Previous studies have established that conformer ensemble learning presents unique challenges: explicit set encoding can improve performance but makes training more challenging due to computational complexity, and model performance often shows strong task dependencies [30, 51]. Despite these challenges, SPiCE consistently

achieves superior performance compared to baseline methods, **outperforming in 34 out of 36 total experimental configurations across all 9 tasks and 4 base models** and often surpassing the state-of-the-art model ConAN-FGW.

Notably, SPiCE demonstrates robust performance across datasets of varying sizes, from the smaller Kraken to the larger Drugs-7.5K and CoV2 datasets, suggesting that our architecture effectively balances computational efficiency with modeling capacity. The improvements are particularly significant with PaiNN, though performance gains vary across tasks, which is consistent with past observation in conformer ensemble modeling where different structural features may dominate different properties.

**Regression tasks.** On the Drugs-7.5K and Kraken datasets, SPiCE demonstrates consistent improvements across all backbone architectures. For Drugs-7.5K, SPiCE achieves relative MAE reductions of 2.79%, 7.89%, and 5.78% for IP, EA, and $\chi$ respectively, compared to the second-best strategy. This is due to its strong geometric feature extraction capabilities complementing our selective information processing. The mid-sized Kraken dataset shows even more improvements, with MAE reductions of 3.66%, 8.78%, 7.61%, and 3.53% across its four targets ($B_5$, L, $BurB_5$, and BurL). The most significant improvement is observed with ClofNet on $BurB_5$, where MAE reduces from 0.5375 to 0.4903. It demonstrates that SPiCE can effectively enhance even simple yet powerful equivariant backbones, a pattern we also observe in classification results.

**Classification tasks.** SPiCE shows strong performance on our highly imbalanced classification tasks. On the moderately imbalanced CoV2-3CL dataset ($\sim$1:10 positive-negative ratio), the PaiNN backbone achieves the most notable improvements, with ROC-AUC increasing by 10.95% and reaching perfect precision. The more challenging CoV2 dataset ($\sim$1:60 ratio) shows similar trends, increasing ROC-AUC by 8.41% and thus highlighting the effectiveness of SPiCE and its robustness to extreme class imbalance.

These comprehensive results highlight the effectiveness of our geometry-aware interaction networks in selectively integrating geometric and topological molecular information. The consistent improvements across diverse tasks, backbones, and data distributions suggest that SPiCE successfully captures meaningful patterns in conformer ensembles.

## 4.3 Analysis of Model Scaling (RQ2)

Processing conformer ensembles introduces significant computational overhead, making it essential to understand how model performance scales with dataset size. To investigate the scalability of SPiCE, we conduct experiments on the EA task using the PaiNN backbone. Starting with the full Drugs-75K dataset (75,099 molecules, 558,002 conformers) [51], we create random subsets ranging from 10% to 100% of the data at 10% intervals.

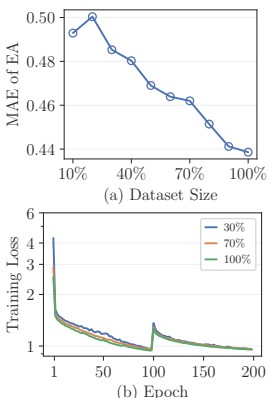

Figure 2: (a) Performance as the dataset size increases from 7.5K to 75K molecules. (b) Training loss trajectories of the first 200 epochs for 30%, 70%, and 100% of 75K molecules.

Our analysis reveals two findings. First, SPiCE shows consistent improvements with increasing data size, reducing MAE from 0.4929 (7.5K molecules) to 0.4386 (75K molecules). As shown in Figure 2(a), the approximately linear scaling relationship suggests that our geometry-aware architecture effectively leverages additional training data without encountering performance plateaus that often characterize capacity-limited models. Second, we plot the loss trajectories at 30%, 70%, and 100% dataset sizes for the first 200 epochs in Figure 2(b). They all show similar patterns despite varying data scales. This consistent optimization behavior suggests that SPiCE maintains stable learning characteristics across different dataset sizes, making it suitable for applications with varied data.

## 4.4 Ablation Studies (RQ3)

To better understand the key design choices, we conducted extensive ablation studies using PaiNN as the backbone and EA as the task. We disabled or modified individual components of SPiCE while keeping other parts unchanged. All experiments used the same protocol to ensure fair comparison.

Figure 3 presents our ablation results. First, the type-1 MoE component for vector feature processing demonstrates the highest importance, with its removal causing a substantial performance drop. This shows the necessity of dedicated equivariant feature handling. Similarly, the topological aggregator proves crucial, confirming that molecular topology provides essential information for conformer ensemble representations. Third, the router mechanisms, both Gumbel sampling and nonlinear activation, significantly influence model performance, indicating that sophisticated expert selection mechanisms are fundamental for effective specialization. Lastly, the gated aggregation component validates the selective cross-conformer integration, while the relatively smaller impact of atom-wise routing and layer-wise MoE suggests some implementation flexibility in these architectural aspects.

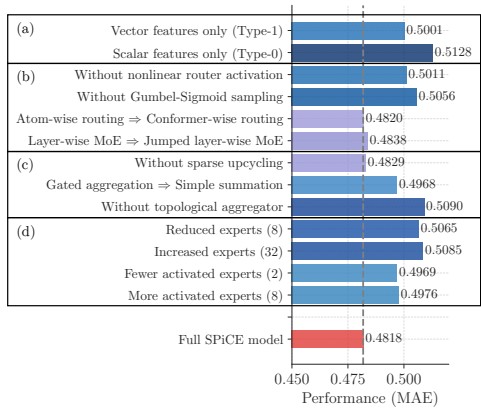

Figure 3: Results of ablation studies regarding (a) feature processing, (b) routing mechanism, (c) training & integration, and (d) expert granularity.

## 4.5 Additional Experiments

We conducted several extra experiments to further validate our approach. Ablation studies on MoE positioning and router z-loss regularization support our architectural choices. The experiments show that conformer-wise routing with layer-wise MoE placement achieves optimal performance, and moderate z-loss regularization ($\lambda = 1e^{-4}$) effectively balances expert utilization with training stability. Additionally, we evaluated SPiCE with invariant backbones and on the BDE dataset [51]. These additional experiments are documented in Appendices G and H.

## 5 Conclusions

We present SPiCE, a hierarchical framework for molecular conformer ensemble learning that combines three key architectural innovations: (1) shared conformer encoding for geometry-preserving molecular processing, (2) geometric mixture-of-experts for specialized handling of scalar and vector features, and (3) hierarchical ensemble encoding that integrates molecular topology with selective cross-conformer communication. SPiCE maintains essential symmetries, both in conformer permutation and geometric transformations. Our comprehensive evaluation demonstrates the effectiveness of SPiCE across diverse molecular prediction tasks.

## Acknowledgments

This work was partially supported by NSF Center for Computer Assisted Synthesis (2202693), National Artificial Intelligence Research Resource (NAIRR) Pilot (240280, 240443), National Science Foundation (2106859, 2211557, 2119643, 2200274, 2303037, 2312501), National Institutes of Health (U54HG012517, U24DK097771, U54OD036472), NEC, Optum AI, SRC JUMP 2.0 Center, Amazon Research Awards, and Snapchat Gifts.

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

# Supplementary Material for **SPiCE**

## A  Related Work

### A.1  Geometric Graph Neural Networks

Message-passing neural networks have been widely adopted in modeling atomistic systems, where nodes represent atoms and edges represent chemical bonds [62]. When incorporating 3D geometric information, these models can be broadly categorized into invariant and equivariant architectures, based on how they handle geometric transformations.

Invariant models satisfy the condition:

$$\phi(g \cdot G) = \phi(G), \quad \forall g \in \mathrm{E}(3)/\mathrm{SE}(3),$$

These models transform equivariant coordinates $\boldsymbol{X}$ into invariant scalar features (type-0) that remain unchanged under Euclidean transformations. Common invariant features include pairwise distances [10], triplet angles [11], torsion angles [12], etc.

Equivariant models, which satisfy:

$$\phi(g \cdot G) = g \cdot \phi(G), \quad \forall g \in \mathrm{E}(3)/\mathrm{SE}(3),$$

can be further divided into two subcategories. Cartesian equivariant models (e.g., EGNN [16], PaiNN [37]) operate directly in Cartesian coordinates, processing scalar (type-0) and vector (type-1)

features while preserving equivariance through restricted operations. In contrast, spherical equivariant models (e.g., SEGNN [63], Equiformer [34, 64]) convert geometric information into steerable features using spherical harmonics, process rotations through Wigner-D matrices, and combine features using Clebsch-Gordan tensor products. This formulation naturally extends to higher-order geometric features, making it particularly effective for capturing complex geometric patterns in molecular structures.

Readers of interest may refer to Duval et al. [65] for a complete review of geometric graph neural networks for 3D atomic systems. We would like to remark that our SPiCE framework is agnostic to the choice of geometric GNN architecture, allowing practitioners to select the most suitable backbone for their specific application while maintaining the benefits of our conformer ensemble processing approach.

## A.2   Learning over Sets

Learning permutation-invariant functions over sets is a fundamental problem in machine learning, with approaches broadly falling into several categories: direct permutation invariance, sorting-based methods, and approximate invariance.

**Direct permutation invariance.** Deep Sets [33] introduced a simple yet powerful architecture where a permutation-sensitive network processes each element independently, followed by a permutation-invariant aggregation operation. Set Transformers [66] extend this using attention mechanisms for more expressive set processing. DSSNet [32] generalizes this framework to handle symmetries beyond permutation invariance. They show that when working with symmetries that combine permutations with element-wise transformations, the processing can be decomposed into two parts: one that handles individual elements and another that processes information aggregated across the entire set. This framework naturally fits our molecular setting, where we need to maintain both the permutation invariance of conformer ensembles and the geometric symmetries of individual conformers. More sophisticated approaches like Janossy pooling [67] capture higher-order interactions by processing subsets of elements together. Self-attention mechanisms [59] offer another perspective, effectively comparing pairs of elements through learned relationships between queries and keys.

**Sorting-based methods.** Another strategy achieves permutation invariance by first arranging elements into a canonical order. Recent advances include methods for learning the sorting function itself [68] and approaches that sort based on learned features [69]. While these methods are computationally efficient, they face challenges in training due to the discrete nature of sorting operations.

**Approximate invariance.** Some methods trade exact invariance for computational efficiency. These include stochastic approaches that sample random permutations [67] and adversarial training techniques that encourage approximate invariance [70].

Our work, while informed by these approaches, focuses on maintaining exact invariance through the principled decomposition provided by DSSNet [32]. We propose a novel type-separated Mixture-of-Experts mechanism to enable selective information processing while achieving the required symmetry.

## A.3   Mixture-of-Experts

When processing molecular conformers, certain atoms or structural features may be more relevant than others for specific properties. Graph pooling approaches like SAGPool achieve this selective processing by focusing computation on important nodes through attention mechanisms. This naturally extends to Mixture-of-Experts (MoE) architectures that employ multiple specialized neural networks with a learned routing mechanism.

The concept of MoE originated from adaptive mixtures of local experts [41], where separate networks handle different subsets of training cases. Recent developments have transformed MoEs from standalone models to components within deeper architectures [71], enabling selective computation at various scales. Modern MoE architectures have demonstrated remarkable success in large-scale models [43, 44, 46, 48, 72], achieving improved data efficiency and computational performance through sparse activation patterns.

Key design considerations of MoE include balanced expert utilization through load balancing, router design, sparse gating for computational efficiency, and appropriate capacity factors. Router z-loss [46] penalizes extreme routing decisions to prevent expert collapse and improve training stability. Expert capacity factors control the maximum number of tokens per expert, balancing computational efficiency with model performance [73]. While traditional MoEs use token-to-expert routing, Expert Choice routing [74] reverses this by allowing experts to select tokens, improving load balancing and computational efficiency. Recent innovations include parameter-efficient MoE variants [75], and Soft MoE [76], which replaces discrete expert assignment with differentiable soft routing to address training instability and scaling limitations. These developments collectively enable more efficient and stable MoE architectures while preserving the benefits of specialized computation.

This MoE approach is suitable for conformer ensemble modeling as experts can specialize in different geometric patterns or conformational states. We further leverage a gated aggregation mechanism which can adaptively weigh the importance of different conformers based on both local geometric features and global molecular context.

# B  PyTorch-like Pseudocode

A PyTorch-like pseudocode is given in Algorithm S1.

# C  Theoretical Details

## C.1  $S_n$–equivariant Linear Layer Structure

**Theorem 1.** *Consider $S_n$–equivariant linear layer $L$ that also respects $G$–symmetry.*

1. *If $L$ takes as input $G$–equivariant features $\mathbb{R}^{n \times 3 \times d}$, then $L(\boldsymbol{X})_i = L_0(\boldsymbol{x}_i)$ operates on each conformer $i$ separately;*

2. *If $L$ takes as input $G$–invariant features $\mathbb{R}^{n \times 1 \times d}$, then $L$ can be decomposed as $L(\boldsymbol{X})_i = L_1(\boldsymbol{x}_i) + L_2\left(\sum_{j=1}^n \boldsymbol{x}_j\right)$, i.e. a local interaction module and a global aggregation module.*

*Proof of Theorem 1.* For the first claim, we begin by noting that any linear function of equivariant features, if it respects $G$–symmetry, must be a $G$–equivariant linear function, hence the output $L(\boldsymbol{X})$ is also $G$–equivariant, and operates on the 3 coordinate channels of the second dimension of $L$ simultaneously. Next we decompose $L$ into separate components from each individual conformer:

$$L(\boldsymbol{X})_i = \sum_{j=1}^n L_{i,j}(\boldsymbol{x}_j), \tag{S1}$$

where each $L_{i,j}$ must be a $G$–equivariant function. Consider the transformation $g = (g_1, \cdots, g_n) \in G^n$ on input $\boldsymbol{X}$: $g \circ \boldsymbol{X} = [g_1 \circ \boldsymbol{x}_1, \cdots, g_n \circ \boldsymbol{x}_n]$, where $g_1, \cdots, g_n$ are independent transformations in $G$. From the $G^n$–equivariance constraint, we have for any $g \in G^n$ that

$$L(g \circ \boldsymbol{X}) = g \circ L(\boldsymbol{X}) \Leftrightarrow \sum_{j=1}^n L_{i,j}(g_j \circ \boldsymbol{x}_j) = g_i \circ \sum_{j=1}^n L_{i,j}(\boldsymbol{x}_j), \forall i$$

By fixing $g_i$ and varying $g_j$ for all $j \neq i$, we see the right hand side remains constant, while each $L_{i,j}(g_j \circ \boldsymbol{x}_j) = g_j \circ L_{i,j}(\boldsymbol{x}_j)$ on the left hand side individually changes with different $g_j$'s. Thus the equation can only hold for any $g$ if $L_{i,j}(\boldsymbol{x}_j) = \boldsymbol{0}$ for any $i \neq j$, i.e. there are no cross-conformer interactions, and consequently (S1) becomes

$$L(\boldsymbol{X})_i = L_{i,i}(\boldsymbol{x}_i).$$

From here we finally consider $S_n$–equivariance: for a permutation $p \in S_n$, defined by $p : [x_i]_{i=1}^n \mapsto [x_{p(i)}]_{i=1}^n$, we have

$$p \circ L(\boldsymbol{X}) = L(p \circ \boldsymbol{X}) \Leftrightarrow L_{p(i),p(i)}(\boldsymbol{x}_{p(i)}) = L_{i,i}(\boldsymbol{x}_{p(i)}), \forall i,$$

which suggests by the arbitrariness of $p$ that $L_{i,i} = L_{j,j}, \forall i, j$. Writing $L_0 = L_{i,i}$ finishes the proof.

**Algorithm S1** Pseudocode of SPiCE in a PyTorch-like style.

```
1   # Inputs:
2   # batch = [z, c, atom_idx]
3   # z : (Nx1) atomic numbers for all M conformers
4   # c : (Nx3) atomic coordinates for all M conformers
5   # atom_idx : (Nx1) index mapping each atom to its molecule ID
6   # conf_idx : (Nx1) index mapping each atom to its conformer ID
7
8   # --- 1) Preprocess ---
9   # build scalar & vector features and auxiliary graph attributes
10  x_s, x_v, 3d_aux = 3d_gnn.preprocess(batch)
11  _, 2d_aux = GIN.preprocess(batch)
12
13  # --- 2) Interaction block ---
14  for i in range(num_blocks):
15      # type-0 feature x_s: Nx1xC
16      # type-1 feature x_v: Nx3xC
17      h_s, h_v, 3d_aux = 3d_gnn[i].forward(x_s, x_v, 3d_aux)
18
19      # normalized type-0 feature h_s: Nx1xC
20      # normalized type-1 feature h_v: Nx3xC
21      h_s = ELN_s[i](x_s + h_s)
22      h_v = ELN_v[i](x_v + h_v)
23
24      # GMoE block, Eqn. (2)-(8)
25      # moe_x: Nx1xC, moe_v: Nx3xC
26      # router scores: Nx(N_I+N_E)
27      moe_x, moe_v, scores = gmoe_block[i](h_s, h_v)
28
29      # conformer set information sharing
30      # h_bar, h_bar_s: MxC
31      h_bar = scatter(moe_x, conf_idx, dim=0, reduce='mean')
32      h_bar_s = GIN[i](h_bar, 2d_aux)
33
34      # gated aggregation, Eqn. (10)-(11)
35      # rev_idx: mapping from conformer-level back to atom ordering
36      x_g = gated_aggr[i](h_s, h_bar_s[rev_idx])
37      x_g = x_g.unsqueeze(1) # x_g: Nx1xC
38
39      # feature update
40      x_s = x_f
41      x_v = moe_v
42
43  # --- 3) Postprocess ---
44  # convert final node-level features to molecule-level outputs
45  out = 3DGNN.postprocess(x_s, x_v, 3d_aux, atom_idx)
```

For the second claim, clearly the output of any $G$–invariant function is also $G$–invariant, and cannot contain any type-1 features. We similarly decompose $L(\boldsymbol{X})$ as

$$L(\boldsymbol{X})_i = \sum_{j=1}^{n} L_{i,j}(\boldsymbol{x}_j), \tag{S2}$$

where each $L_{i,j}$ is an $G$–invariant linear function. Consider $p \in S_n$ again, we have

$$p \circ L(\boldsymbol{X}) = L(p \circ \boldsymbol{X}) \Leftrightarrow L(\boldsymbol{X})_{p(i)} = L(p \circ \boldsymbol{X})_i, \forall i$$

$$\Leftrightarrow \sum_{j=1}^{n} L_{p(i),j}(\boldsymbol{x}_j) = \sum_{j=1}^{n} L_{i,j}(\boldsymbol{x}_{p(j)}), \forall i$$

$$\Leftrightarrow \sum_{j=1}^{n} L_{p(i),j}(\boldsymbol{x}_j) = \sum_{k=1}^{n} L_{i,p^{-1}(k)}(\boldsymbol{x}_k), \forall i$$

$$\Leftrightarrow \sum_{j=1}^{n} \left[ L_{p(i),j}(\boldsymbol{x}_j) - L_{i,p^{-1}(j)}(\boldsymbol{x}_j) \right] = 0, \forall i,$$

where in the third equivalence we used $k = p(j)$ to rewrite the summation. From the independent variabilities of $\boldsymbol{x}_i$, each member of the above summation must take value 0, in other words

$$L_{p(i),j} = L_{i,p^{-1}(j)}, \forall i, j, p,$$

which is equivalent to

$$L_{p(i),p(k)} = L_{i,k}, \forall i, k, p,$$

via substitution of $k = p^{-1}(j)$. This means:

- $L_{i,i} = L_{j,j} = L_{\mathrm{e}}, \forall i, j$;

- $L_{i_1,j_1} = L_{i_2,j_2} = L_{\mathrm{n}}, \forall i_1 \neq j_1, i_2 \neq j_2$;

and therefore (S2) becomes

$$L(\boldsymbol{X})_i = L_{\mathrm{e}}(\boldsymbol{x}_i) + \sum_{j \neq i} L_{\mathrm{n}}(\boldsymbol{x}_j)$$

$$= (L_{\mathrm{e}} - L_{\mathrm{n}})(\boldsymbol{x}_i) + L_{\mathrm{n}}\left( \sum_{j=1}^{n} \boldsymbol{x}_j \right),$$

where in the second equality we used the fact that the summation of a shared linear mapping of components is equal to the same linear mapping of the summation of components. Taking $L_1 = L_{\mathrm{e}} - L_{\mathrm{n}}$ and $L_2 = L_{\mathrm{n}}$ finishes the proof. $\qquad\square$

### C.2  $S_n$–equivariance of Cross-Attention Output $\boldsymbol{X}^{\mathrm{s}}$

Here we elaborate on the symmetry preservation of the gated aggregation operation discussed in Section 3.2.4, and prove the $S_n$–equivariance of the attention output. Looking at (10) and (11), notice the input $\mathbf{H}^{\mathrm{s}}$ is $S_n$–equivariant while $\bar{\mathbf{H}}^{\mathrm{s}}$ is $S_n$–invariant, therefore for a permutation $p \in S_n$,

$$\mathrm{Attn}(p \circ \mathbf{H}^{\mathrm{s}}, p \circ \bar{\mathbf{H}}^{\mathrm{s}}) = \mathrm{Attn}(p \circ \mathbf{H}^{\mathrm{s}}, \bar{\mathbf{H}}^{\mathrm{s}}) = \mathrm{softmax}\left( \frac{(p \circ \mathbf{H}^{\mathrm{s}} \boldsymbol{W}_Q)(\bar{\mathbf{H}}^{\mathrm{s}} \boldsymbol{W}_K)^{\mathsf{T}}}{\sqrt{d_k}} \right) \bar{\mathbf{H}}^{\mathrm{s}} \boldsymbol{W}_V$$

$$= p \circ \mathrm{softmax}\left( \frac{(\mathbf{H}^{\mathrm{s}} \boldsymbol{W}_Q)(\bar{\mathbf{H}}^{\mathrm{s}} \boldsymbol{W}_K)^{\mathsf{T}}}{\sqrt{d_k}} \right) \bar{\mathbf{H}}^{\mathrm{s}} \boldsymbol{W}_V = p \circ \mathrm{Attn}(\mathbf{H}^{\mathrm{s}}, \bar{\mathbf{H}}^{\mathrm{s}}), \quad (S3)$$

where the third equality holds since individual operations on each conformer preserves permutation equivariance. This proves the overall $S_n$–equivariance of the attention module.

Table S1: Hyperparameters for each backbone model on dataset Drugs-7.5K and Kraken.

| Dataset | Backbone | Epochs | Batch | LR | Patience | Experts | Act. Experts | Upcycle | $\tau$ | $\beta$ |
|---|---|---|---|---|---|---|---|---|---|---|
| Drugs-7.5K | PaiNN | 2000 | 32 | 2e-4 | 400 | 8 | 2 | 100 | 0.1 | 1e-3 |
| | ViSNet | 2000 | 32 | 1.5e-4 | 600 | 8 | 4 | 50 | 0.1 | 1e-3 |
| | ClofNet | 2000 | 32 | 1.5e-4 | 400 | 8 | 4 | 50 | 0.1 | 1e-3 |
| | Equiformer | 2000 | 32 | 2e-4 | 400 | 8 | 4 | 50 | 0.1 | 1e-3 |
| Kraken | PaiNN | 2000 | 16 | 3e-4 | 400 | 16 | 2 | 50 | 0.1 | 1e-4 |
| | ViSNet | 2000 | 32 | 4e-4 | 400 | 16 | 4 | 100 | 1.0 | 0.1 |
| | ClofNet | 2000 | 16 | 1e-4 | 400 | 8 | 2 | 100 | 1.5 | 1e-3 |
| | Equiformer | 2000 | 8 | 3e-4 | 400 | 16 | 2 | 50 | 1.0 | 1e-4 |

## D  Details of Experimental Settings

Different backbones have varying architecture complexities and capacities to learn molecular representations. Datasets also differ in size and complexity of the prediction task. Hyperparameter tuning for each backbone-dataset pair ensures optimal performance by balancing model capacity, data size, and task difficulty. For example, larger datasets like Kraken may require more experts and epochs to sufficiently learn, while simpler backbones like ClofNet need fewer epochs on Drugs-7.5K. The hyperparameters include number of experts, number of activated experts, Gumbel-Softmax sampling temperature, auxiliary z-loss weight, and upcycling epochs.

For regression tasks, each backbone model with a different aggregation strategy is trained with the same set of hyperparameters, optimized through 20 iterations of Bayesian Optimization. Specific settings are summarized in Table S1 for regression datasets. For classification tasks, all experiments use the same settings as PaiNN on Drugs-7.5K, as classification is less sensitive to hyperparameters than regression and Drugs-7.5K is a reliable molecular property prediction benchmark.

## E  Details of Datasets

We evaluate our method on four diverse molecular datasets. Their statistics is summarized in Table S2.

**Drugs-7.5K.**  A subset of 7,500 molecules downsampled from GEOM-Drugs dataset, with three quantum mechanical properties:

- Ionization Potential (IP): The energy required to remove an electron from a neutral molecule ($IP = E_{cation} - E_{neutral}$).

- Electron Affinity (EA): The energy change when adding an electron ($EA = E_{neutral} - E_{anion}$).

- Electronegativity ($\chi$): Measuring electron attraction tendency ($\chi = -\partial E/\partial N$).

**Kraken.**  A dataset of 1,552 monodentate organophosphorus(III) ligands with DFT-computed conformer ensembles. We focus on four 3D steric descriptors (measured in Å): Sterimol $B_5$, Sterimol L, buried Sterimol $B_5$, and buried Sterimol L, which are crucial for QSAR modeling in catalyst design.

**CoV2 and CoV2-3CL.**  Two highly imbalanced classification datasets derived from GEOM-Drugs, containing experimental data for SARS-CoV-2 inhibition:

- CoV2-3CL: Tests specific inhibition of the SARS-CoV-2 3CL protease.

- CoV2: Evaluates general SARS-CoV-2 inhibition in human cell assays.

The class imbalance is summarized in Table S2b, where positive samples (hits) represent only a small fraction of the total molecules.

Table S2: Dataset statistics for all four datasets and dataset splits for COVID-related datasets.

| Dataset | #Molecules | #Conformers | #Heavy atoms | #Rot. bonds |
|---|---|---|---|---|
| Drugs-7.5K | 7,509 | 54,202 | 30.67 | 7.45 |
| Kraken | 1,552 | 21,287 | 23.70 | 9.05 |
| CoV2 | 5,466 | 72,744 | 24.57 | 4.83 |
| CoV2-3CL | 755 | 7,742 | 14.51 | 2.41 |

| Split | CoV2-3CL | CoV2 |
|---|---|---|
| Train | 50 (485) | 53 (3,294) |
| Validation | 15 (157) | 17 (1,096) |
| Test | 11 (162) | 22 (1,086) |
| Total | 76 (804) | 92 (5,476) |

(a) Molecular composition statistics. Numbers of heavy atoms and rotatable bonds ("rot. bonds") are averaged per molecule.

(b) Dataset partitioning showing number of active compounds (total compounds in parentheses)

## F    Details of Baselines

We evaluate SPiCE against two categories of baselines: ensemble learning methods with explicit set encoders and specialized conformer ensemble models.

**Ensemble learning with set encoders.**    We implement our framework with four representative 3D backbones that guarantee E(3) or SE(3) equivariance: PaiNN [37], ViSNet [58], ClofNet [13], and Equiformer [34]. For each backbone, we compare GAIN against three conformer aggregation strategies following Zhu et al. [51]:

- Mean pooling: The simplest approach that computes the average of conformer embeddings $\{h_i\}_{i=1}^n$ generated by 3D GNNs:

$$h_{\text{MEAN}} = \frac{1}{n} \sum_{i=1}^{n} h_i. \tag{S4}$$

- DeepSets [33]: A permutation-invariant function that processes the ensemble through a Multi-Layer Perceptron (MLP) $\phi$, followed by sum pooling and another MLP $\rho$:

$$h_{\text{DS}} = \text{MLP}_\rho \left( \sum_{i=1}^{n} \text{MLP}_\phi(h_i) \right), \tag{S5}$$

where $\text{MLP}_\phi$ transforms individual embeddings and $\text{MLP}_\rho$ processes the aggregated features. This approach preserves more individual conformer information than mean pooling at the cost of additional non-linear transformations.

- Self-attention [59]: Computes a weighted sum of embeddings using attention scores:

$$h_{\text{ATTN}} = \sum_{i=1}^{n} \alpha_i h_i, \tag{S6}$$

$$\alpha_i = \text{softmax}(h_i^\mathsf{T} W h_i). \tag{S7}$$

This approach captures pairwise interactions between conformers through learned attention weights.

After obtaining the ensemble embeddings through these set encoders, a linear projection head generates the final predictions.

**Specialized conformer ensemble models.**    We also compare against two state-of-the-art models specifically designed for conformer ensemble learning:

- ConfNet [60]: Extends DSSNet [32] as an explicit set encoder, applying permutation-invariant operations directly to the conformer ensemble while maintaining geometric equivariance.

- ConAN-FGW [31]: Introduces a novel 2D–3D aggregation mechanism based on the Fused Gromov-Wasserstein Barycenter problem. It combines this with efficient conformer generation using distance geometry through RDKit, enabling joint optimization of conformer generation and property prediction.

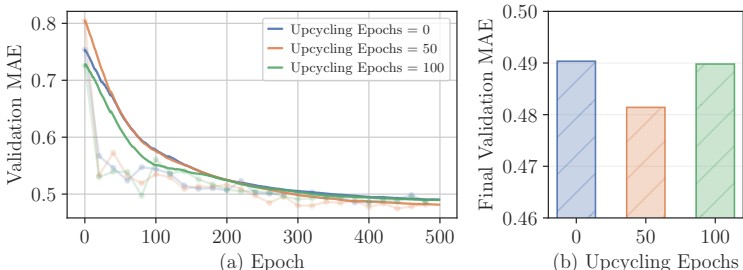

Figure S1: Impact of upcycling epochs on model performance. (a) Validation loss curves for different upcycling epochs on the Drugs-7.5K dataset using ViSNet backbone for EA prediction. (b) MSE comparison of validation errors.

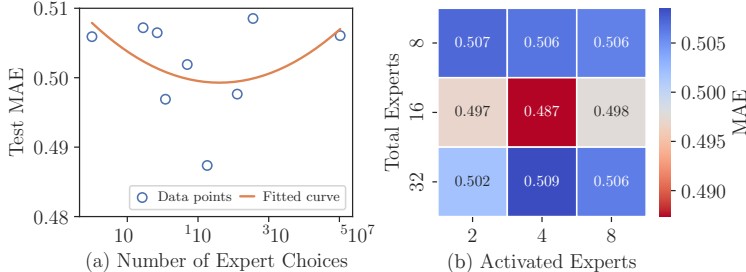

Figure S2: Analysis of expert combinations on model performance. Results show how varying the number of total experts (8–32) and active experts (2–8) affects prediction accuracy on the Drugs-7.5K dataset using PaiNN backbone for EA prediction.

## G  Detailed Ablation Studies

To better understand the key design choices in SPiCE, we conduct comprehensive ablation studies examining four critical aspects: (1) the effectiveness of sparse upcycling for stabilizing early-stage training, (2) the impact of expert granularity on model performance, (3) the optimal positioning of TS-MoE within the architecture, and (4) the influence of router z-loss on training stability and expert utilization. Here we present analyses of the first two aspects, while full results for the rest are provided in Appendix G.

**Sparse upcycling.** Training both GNN representation and MoE components simultaneously can be challenging in early stages. The sparse upcycling strategy [49] addresses this by separating the training into two phases: first training a single expert to optimize representation learning, then cloning it into multiple experts with a newly initialized router for specialization.

As shown in Figure S1, we evaluate different upcycling epochs (0, 50, 100) on the Drugs-7.5K dataset using ViSNet for EA prediction. Results indicate that 50 upcycling epochs yield optimal performance, while excessive upcycling epochs can slow convergence. This confirms that appropriate upcycling improves both convergence speed and final accuracy.

**Expert granularity.** While finer expert granularity can increase model flexibility through more possible combinations (e.g., $\binom{8}{2} = 28$ vs. $\binom{16}{4} = 1820$) [73], it also introduces computational overhead and potential training instability. Figure S2 presents a systematic study varying the number of total experts (8, 16, 32) and activated experts (2, 4, 8) on the Drugs-7.5K dataset with PaiNN backbone. The results reveal a U-shaped performance curve, with optimal performance at $\binom{16}{4} = 1820$ combinations. This suggests a sweet spot balancing model flexibility with computational efficiency for molecular conformer ensemble representation.

**MoE positioning.** Table S3a compares different MoE configurations on the Kraken dataset's $BurB_5$ target using PaiNN. Our experiments examine three key design choices: conformer-wise vs. atom-wise routing, layer-wise MoE placement, and non-linear router activation. Results show that conformer-wise routing with layer-wise MoE and softmax activation achieves the best performance, supporting our design choice of treating conformers rather than atoms as routing objectives.

Table S3: Ablation studies on (a) different architectural choices, (b) router z-loss weight ($\lambda$). Results show validation MAE on Kraken (BurB$_5$) and Drugs-7.5K (EA) datasets using PaiNN and ViSNet respectively.

| Atom-Wise Routing | Layer-Wise MoE | Router Activation | MAE |
|:---:|:---:|:---:|:---:|
| ✗ | ✓ | ✓ | 0.1645 |
| ✓ | ✗ | ✓ | 0.1515 |
| ✓ | ✓ | ✗ | 0.1546 |
| ✓ | ✓ | ✓ | **0.1470** |

(a) Impact of architectural choices

| $\lambda$ | MAE |
|:---:|:---:|
| $1e^{-6}$ | 0.4973 |
| $1e^{-5}$ | 0.4960 |
| $1e^{-4}$ | **0.4888** |
| $1e^{-3}$ | 0.4985 |

(b) Z-loss $\lambda$

**Router z-loss.** Router z-loss penalizes extreme routing decisions and helps balance expert utilization. Table S3b shows the impact of different z-loss weights ($\lambda$) on validation error using the ViSNet backbone on Drugs-75K. A moderate weight of $1e^{-4}$ achieves optimal performance, suggesting that proper regularization improves model stability without overly constraining expert specialization. Larger weights, while promoting more balanced expert utilization, can lead to training instability due to excessive penalization.

# H  Additional Experiments

**BDE dataset.** To showcase our method is effective and scalable to reaction-based prediction task, we conducted experiments on the BDE dataset [77]. The BDE dataset comprises 5,915 organometallic catalysts ($ML_1L_2$) with diverse ligands and metal centers, including DFT-computed binding energies for conformer ensembles of unbound and bound states. Conformers are generated via Open Babel [78] and force field optimizations, approximating global minima. Due to the high cost of DFT optimization, precise conformer ensembles are generally unknown at inference, making this a realistic and challenging benchmark. The goal is to predict binding energies from either individual or ensemble conformers of the catalyst in both states.

Different from the experimental settings in the main paper, this task involves two distinct sets of conformers as input, corresponding to the unbound and bound states of the catalyst. To accommodate this, we employ two parameter-independent copies of SPiCE, with their regression heads removed, to separately process each conformer set. The resulting molecular-level features are then concatenated and passed through a regression head to predict the final binding energy. Table S4a presents the performance of SPiCE on this dataset in comparison with the two other ensemble models Mean and DeepSets with PaiNN [37] and Equiformer [34] as backbones. It is seen that SPiCE consistently outperforms the baselines, demonstrating its effectiveness in reaction-based prediction tasks.

**Invariant Type-0-only model.** Our SPiCE framework is designed as a plug-and-play architecture that can be adapted to work with different geometric neural network backbones. While our method

Table S4: Additional experiments on the BDE dataset and invariant models. **Bold** values indicate best performance.

| Backbone | Model | Binding Energy |
|:---:|:---:|:---:|
| PaiNN [37] | Mean | 1.8744 |
| | DeepSets | 1.9164 |
| | SPiCE | **1.8528** |
| Equiformer [34] | Mean | 1.9136 |
| | DeepSets | 1.9540 |
| | SPiCE | **1.8978** |

| Model | B$_5$ | L |
|:---:|:---:|:---:|
| Mean | 0.3172 | 0.4258 |
| DeepSets | 0.2627 | 0.3777 |
| SPiCE (Invariant) | **0.2446** | **0.3379** |

(a) Reaction-based regression task (BDE) in MAE ($\downarrow$).

(b) Invariant type-0-only model on Kraken dataset with DimeNet++ [79] as backbone.

can accommodate both invariant and equivariant backbones, we primarily focus on equivariant 3D GNNs (PaiNN, ClofNet, Equiformer, and VisNet) due to their superior representational capabilities. Equivariant features have been shown to provide steeper learning curves, improved data efficiency, and finer angular resolution for capturing molecular structural details [15, 80, 81].

To demonstrate the versatility of SPiCE and address potential questions about backbone compatibility, we developed a simplified variant that operates exclusively on invariant (type-0) features, compatible with invariant-only backbones such as DimeNet++ [79]. This adaptation removes the type-1 processing branch including vector features, type-1 router, and the corresponding expert network to accommodate scalar-only representations.

We evaluate this invariant variant on the Kraken dataset against strong baselines including DeepSets and mean pooling (results in Table S4b). We note that this experiment is not intended as a direct comparison between invariant and equivariant approaches. Rather, it validates the effectiveness of our design in a constrained setting and confirms the adaptability of our model across different architectural paradigms. Even without equivariant information, our adapted framework can outperform the invariant baselines, demonstrating that the core design remains beneficial across different geometric representations.

# I    Code and Dataset Availability

The implementation of this work can be found in this repository: https://github.com/DannieS YD/SPiCE.

# J    Limitations and Future Work

While SPiCE achieves strong performance across a variety of molecular property prediction tasks, the hierarchical architecture and dependence on equivariant GNN backbones introduce additional computational overhead compared to simpler pooling methods, potentially limiting scalability in resource-constrained environments or applications requiring real-time inference. Furthermore, although SPiCE facilitates cross-conformer interactions through attention-based integration, the framework does not explicitly incorporate thermodynamic priors or statistical mechanical principles that govern conformational ensembles.

Future work could address these limitations through several directions: developing more efficient attention mechanisms or approximation strategies to reduce computational cost, incorporating physical distribution information, and exploring hybrid approaches that balance computational efficiency with physical grounding.

# K    Raw Performance Data

The raw performance data with standard deviation of Table 1 is summarized in Table S5.

Table S5: Raw performance (mean ± standard deviation) in terms of MAE (↓) for seven regression tasks (Drugs-7.5K, Kraken) and ROC scores (↑) for two classification tasks (CoV2, CoV2-3CL).

| Backbone | Ensemble Strategy | Drugs-7.5K (MAE, ↓) | | | Kraken (MAE, ↓) | | | | CoV2 | 3CL |
|---|---|---|---|---|---|---|---|---|---|---|
| | | IP | EA | $\chi$ | $B_5$ | L | $BurB_5$ | BurL | ROC (↑) | ROC (↑) |
| PaiNN [37] | Mean | $0.5410_{\pm0.0462}$ | $0.4966_{\pm0.0336}$ | $0.2963_{\pm0.0190}$ | $0.2877_{\pm0.0252}$ | $0.3950_{\pm0.0233}$ | $0.1817_{\pm0.0091}$ | $0.1472_{\pm0.0039}$ | $0.5722_{\pm0.0518}$ | $0.8850_{\pm0.1209}$ |
| | DeepSets | $0.5396_{\pm0.0534}$ | $0.5091_{\pm0.0129}$ | $0.2982_{\pm0.0052}$ | $0.2225_{\pm0.0218}$ | $0.3619_{\pm0.0192}$ | $0.1693_{\pm0.0111}$ | $0.1324_{\pm0.0091}$ | $0.5802_{\pm0.0356}$ | $0.6808_{\pm0.1239}$ |
| | Attention | $0.6318_{\pm0.0327}$ | $0.5985_{\pm0.0160}$ | $0.3488_{\pm0.0126}$ | $0.3496_{\pm0.0140}$ | $0.4109_{\pm0.0167}$ | $0.2123_{\pm0.0005}$ | $0.1506_{\pm0.0029}$ | $0.4179_{\pm0.0357}$ | $0.6984_{\pm0.1994}$ |
| | SPiCE | $0.5281_{\pm0.0666}$ | $0.4929_{\pm0.0331}$ | $0.2792_{\pm0.0030}$ | $0.2178_{\pm0.0376}$ | $0.3548_{\pm0.0199}$ | $0.1564_{\pm0.0154}$ | $0.1292_{\pm0.0031}$ | $0.5910_{\pm0.0831}$ | $0.8880_{\pm0.2113}$ |
| ClofNet [13] | Mean | $0.5935_{\pm0.0672}$ | $0.5441_{\pm0.0072}$ | $0.3121_{\pm0.0266}$ | $0.3986_{\pm0.0211}$ | $0.5674_{\pm0.0423}$ | $0.2857_{\pm0.0332}$ | $0.2327_{\pm0.0176}$ | $0.3900_{\pm0.0042}$ | $0.7580_{\pm0.1898}$ |
| | DeepSets | $0.5912_{\pm0.0447}$ | $0.5533_{\pm0.0292}$ | $0.3153_{\pm0.0174}$ | $0.3314_{\pm0.0187}$ | $0.5375_{\pm0.0154}$ | $0.2532_{\pm0.0043}$ | $0.1983_{\pm0.0008}$ | $0.6208_{\pm0.0354}$ | $0.7628_{\pm0.0184}$ |
| | Attention | $0.6694_{\pm0.0264}$ | $0.5949_{\pm0.0352}$ | $0.3578_{\pm0.0096}$ | $0.4979_{\pm0.0199}$ | $0.6118_{\pm0.0328}$ | $0.3353_{\pm0.0109}$ | $0.2502_{\pm0.0099}$ | $0.3707_{\pm0.0149}$ | $0.8182_{\pm0.1042}$ |
| | SPiCE | $0.5747_{\pm0.0362}$ | $0.5283_{\pm0.0186}$ | $0.3059_{\pm0.0035}$ | $0.3193_{\pm0.0234}$ | $0.4903_{\pm0.0311}$ | $0.2477_{\pm0.0113}$ | $0.1913_{\pm0.0098}$ | $0.6730_{\pm0.0347}$ | $1.0000_{\pm0.0972}$ |
| Equiformer [34] | Mean | $0.5457_{\pm0.0349}$ | $0.4932_{\pm0.0125}$ | $0.2977_{\pm0.0160}$ | $0.2303_{\pm0.0059}$ | $0.3830_{\pm0.0291}$ | $0.1680_{\pm0.0004}$ | $0.1259_{\pm0.0011}$ | $0.5601_{\pm0.0351}$ | $0.8387_{\pm0.0982}$ |
| | DeepSets | $0.5404_{\pm0.0247}$ | $0.4888_{\pm0.0154}$ | $0.2990_{\pm0.0016}$ | $0.2564_{\pm0.0159}$ | $0.3772_{\pm0.0008}$ | $0.1782_{\pm0.0120}$ | $0.1234_{\pm0.0023}$ | $0.5125_{\pm0.0346}$ | $0.7134_{\pm0.0194}$ |
| | Attention | $0.5488_{\pm0.0205}$ | $0.4923_{\pm0.0371}$ | $0.2896_{\pm0.0247}$ | $0.3187_{\pm0.0074}$ | $0.4508_{\pm0.0352}$ | $0.1673_{\pm0.0058}$ | $0.1425_{\pm0.0195}$ | $0.3882_{\pm0.0377}$ | $0.7881_{\pm0.0614}$ |
| | SPiCE | $0.5318_{\pm0.0254}$ | $0.4830_{\pm0.0453}$ | $0.2816_{\pm0.0332}$ | $0.2241_{\pm0.0102}$ | $0.3456_{\pm0.0291}$ | $0.1611_{\pm0.0004}$ | $0.1229_{\pm0.0021}$ | $0.5650_{\pm0.0457}$ | $0.8405_{\pm0.0823}$ |
| ViSNet [58] | Mean | $0.5593_{\pm0.0392}$ | $0.4927_{\pm0.0451}$ | $0.2862_{\pm0.0347}$ | $0.2811_{\pm0.0163}$ | $0.3970_{\pm0.0461}$ | $0.1874_{\pm0.0036}$ | $0.1469_{\pm0.0022}$ | $0.6035_{\pm0.0623}$ | $0.7447_{\pm0.0376}$ |
| | DeepSets | $0.5280_{\pm0.0449}$ | $0.4987_{\pm0.0515}$ | $0.2846_{\pm0.0248}$ | $0.3104_{\pm0.0247}$ | $0.4113_{\pm0.0222}$ | $0.1716_{\pm0.0116}$ | $0.1314_{\pm0.0242}$ | $0.6626_{\pm0.0036}$ | $0.4160_{\pm0.0425}$ |
| | Attention | $0.5593_{\pm0.0455}$ | $0.4988_{\pm0.0355}$ | $0.2944_{\pm0.0108}$ | $0.3755_{\pm0.0129}$ | $0.4195_{\pm0.0336}$ | $0.2384_{\pm0.0172}$ | $0.1394_{\pm0.0044}$ | $0.5262_{\pm0.0472}$ | $0.7158_{\pm0.1312}$ |
| | SPiCE | $0.5384_{\pm0.0298}$ | $0.4538_{\pm0.0491}$ | $0.2814_{\pm0.0155}$ | $0.2715_{\pm0.0130}$ | $0.3807_{\pm0.0399}$ | $0.1657_{\pm0.0120}$ | $0.1277_{\pm0.0077}$ | $0.6890_{\pm0.0629}$ | $0.7195_{\pm0.0837}$ |

