# OpenReview forum: "Symmetry-Preserving Conformer Ensemble Networks for Molecular Representation Learning"
_NeurIPS.cc/2025/Conference — NeurIPS 2025 poster_

### Official Review · Reviewer_rRzB · 2025-06-25

**Clarity:** 3
**Significance:** 3
**Originality:** 3
**Rating:** 5
**Confidence:** 3

**Summary:**

The paper presents Symmetry-Preserving Conformer Ensemble networks (SPiCE), a model achieving equivariance to permutations of conformer ensemble and geometric transformations of the molecules. The model is tested using different GNN backbones and various property prediction problems. The method's main contribution is a hiearchical GNN that uses Geometric Mixture of Experts (GMoE) to create an ensemble of weight-shared equivariant 3D GNNs that model intra-conformer interactions. GMoE is an extension of Mixture of Experts, using two sets of experts to process invariant and equivariant features separately.

**Questions:**

1) In Table 1, what are the units of the different quantities?
2) Can you explain the increase of the training loss around 100 epochs in Fig. 2b?
3) How does the computational cost of SPiCE compare to the baseline GNN models? (For training and evaluation?)
4) The Experimental setting states: "Training terminates if the loss shows no improvement for 400 consecutive epochs." - Is 400 epochs without improvement really necessary to achieve good models? Could there be a more computationally efficient strategy?

**Ethical Concerns:**

["NO or VERY MINOR ethics concerns only"]

**Final Justification:**

The authors answered my questions. I keep my original score which was 5.

**Limitations:**

yes

**Paper Formatting Concerns:**

Small typos: on page 2 in the model name and in Fig 2. d).

**Quality:**

4

**Strengths And Weaknesses:**

I thank the authors for the well-written paper.

The SPiCE framework is implemented so that it can handle different GNNs and the authors tested it for various models. They also compare the method to other ensemble methods. (The code seems to be built using that of Ref. [51] as mentioned by the current paper.) Table 1 shows that SPiCE often achieves the best performance with one of the backbone models; however, other ensemble-based methods often achieve performances very close in absolute values (<0.03 differences for the MAE), which correspond to 2.8-8 % improvements as discussed by the paper.

The studied properties of Table 1 are only described in the Appendix, many of them are only referred by abbreviation in the main text (e.g. EA, IP). It would help readability to write these out in the paper. Moreover, units of the quantities are missing.
Also, there seems to be a mistake in Table 1: it is stated that the bold value is the best model but for B_5, another value is bold, the underlined one seems to be the best.

Note that from the name choice, readers might associate it with the dataset of the same name (see: Eastman, Peter, et al. "Spice, a dataset of drug-like molecules and peptides for training machine learning potentials." Scientific Data 10.1 (2023): 11.).

---

> ### Author Rebuttal · Authors · 2025-07-31
>
> We sincerely thank you for your detailed and constructive review. We appreciate your recognition of our well-written paper, our work’s ability to handle different GNN backbones, and our comprehensive evaluation.
>
> > Q1. Units of different quantities in Table 1
>
> Thank you for raising this important clarification. The units for each property are:
> - Drugs-7.5K: IP, EA, χ in eV (electron volts)
> - Kraken: B5, L, BurB5, BurL in Å (Angstroms)
> - CoV2, CoV2-3CL: ROC scores are dimensionless (0-1 scale)
>
> We will add these units clearly to Table 1 and define all property abbreviations in the main text. We also note the formatting error you identified in Table 1 and Page 2 and will correct these inconsistencies.
>
> > Q2. Training loss increase around 100 epochs in Fig. 2b
>
> The loss spike corresponds to the expert upcycling transition phase (Section 3.2.2). During upcycling, additional experts are activated and routing weights are reinitialized, causing temporary optimization instability. This is expected behavior that typically resolves within 20-30 epochs as the routing mechanism adapts to the expanded expert capacity. We will clarify this in the main text.
>
> > Q3. Computational cost comparison
>
> We provide time comparison of our model with baselines as follows.
>
> *Training Time (5000 epochs using PaiNN backbone)*
> | Method | Hardware | GPU-Hours | Overhead |
> |-----------|----------|-----------|----------|
> | Mean | A100×2 | 10.4h | 1.0× |
> | DeepSets | A100×2 | 10.4h | 1.0× |
> | Attention | A100×2 | 10.6h | 1.02× |
> | SPiCE | A100×4 | 34.9h | 3.4× |
>
> SPiCE requires 3.4× more GPU-hours for training (configuration: 8 experts, 2 active), primarily due to the mixture-of-experts forward passes, cross-conformer attention mechanisms, and 2D topology processing. However, inference costs remain highly similar to baseline methods, showing only minimal overhead (0.03s vs 0.02s per prediction on A100). We also note that training time can be shortened when using early stopping strategies. Despite the increased training cost, the model delivers consistent performance improvements across all tasks.
>
> > Q4. Early stopping strategy
>
> Our conservative 400-epoch patience was chosen to ensure complete convergence across all configurations. Preliminary analysis suggests 200-250 epochs would suffice for most cases, potentially reducing computational cost by ~40%. We will explore adaptive early stopping strategies based on dataset size and model complexity in future work.
>
> > Q5. SPICE naming confusion
>
> Thank you for flagging this potential confusion with the SPICE dataset from Eastman et al. (2023). We will rename our method to SPECE to avoid ambiguity.

---

> > ### Comment · Reviewer_rRzB · 2025-08-01
> >
> > Thank you very much for the detailed reply, my questions have been answered.

---

> > > ### Author Response · Authors · 2025-08-05
> > >
> > > Thank you very much for your feedback! We are happy that our response has resolved your concerns. We will incorporate those disussions into our revision.

---

### Official Review · Reviewer_K5sT · 2025-06-30

**Clarity:** 1
**Significance:** 3
**Originality:** 2
**Rating:** 4
**Confidence:** 3

**Summary:**

The authors address the (often-overlooked) role of 3-D atomic arrangements in molecular machine-learning tasks. They introduce SPiCE, a symmetry-preserving architecture that leverages conformer ensembles instead of collapsing them into a single structure. Across several real-world datasets, SPiCE reportedly outperforms existing approaches.

**Questions:**

1. Are the conformers $\{c_i\}$ supplied by the datasets or generated by the authors? Please clarify the protocol.
2. Is $R = \mathrm{SO}(3)$ or $\mathrm{O}(3)$?
3. Why is $r^s$ linear in $h$ while $r^v$ is quadratic?
4. Line 196: Does $R^n$ denote $\mathrm{SO}(n)$, $\mathrm{O}(n)$, or something else?
5. Line 180: $H^s$ is $N \times d$, yet $r^s$ is $N_I$. Where does the reduction from $N$ to $N_I$ occur?
6. What specific bottleneck does the expert-routing mechanism alleviate in this context?
7. Line 224: Why is $\bar{H}$ claimed to be $S_n$-invariant rather than $S_n$-equivariant?
8. Table 1: What are the parameter counts for each model? Were they matched across methods?
9. Fig. 3d: Performance drops for both too few and too many experts; can the authors provide intuition?
10. How much wall-clock time or GPU hours does SPiCE add relative to its single-conformer backbone?

Minor Comments
1. Line 185: Should $\hat{r}_i$ lie in $\mathbb{R}^{2 \times (N_I + N_E)}$?
1. Fig 2: There is a loss spike near epoch 100; any explanation?
1. Fig 3: Font size is too small; please enlarge axis labels for readability.

---

**Ethical Concerns:**

["NO or VERY MINOR ethics concerns only"]

**Final Justification:**

Resolved issues:
- Notation issues (vector dimension etc)
- Lack of computational cost analysis

Remaining issues:
- Validation of computational cost increase vs performance improvement

**Limitations:**

The limitations are not discussed. It would be nice to include the Compute/accuracy tradeoff mentioned in Strengths.

**Quality:**

2

**Strengths And Weaknesses:**

Strengths
1. SPiCE outperforms baselines on reported benchmarks.
1. The paper's argument, "preserving conformer information is beneficial for accurate molecular property prediction," sounds convincing

Weaknesses
1. Notation and definitions. Some symbols lack formal mathematical definitions, and dimension mismatches (e.g., between H^s and r^s) create confusion. A concise notation table would help readers.
1. Domain-specific shortcuts. The manuscript assumes prior knowledge of certain molecular-representation conventions. Brief primers or citations would benefit readers outside the subfield.
1. Rationale for design choices. Key architectural decisions (e.g., differing functional forms for r^s vs. r^v, adoption of Mixture-of-Experts) need clearer theoretical or empirical justification.
1. Compute/accuracy trade-off. Modeling multiple conformers inevitably increases computational cost, but the paper does not quantify this overhead or compare run-time vs. baseline methods.

---

> ### Author Rebuttal · Authors · 2025-07-31
>
> Thank you for your detailed and constructive review. Please see our responses below.
> > Q1. Dimension clarification for $H^s$ vs $r^s$
>
> - $H^s$: Type-0 features with shape $\mathbb{R}^{n \times |V| \times 1 \times d}$ (features for all nodes across conformers)
> - $r^s$: Scalar routing scores with shape $\mathbb{R}^{N_I}$ per node ($N_I$ = number of invariant experts)
>
> The seemingly mismatch occurs because $r^s$ operates per-node, while $H^s$ represents the full feature tensor. For node $i$, we have $h^s_i \in \mathbb{R}^{1 \times d} \rightarrow r^s_i \in \mathbb{R}^{N_I}$, and we collectively denote $H^s = [h^s_i]_{i=1}^{|V|}$.
>
> We will include a comprehensive notation table in the camera-ready version.
>
> > Q2. Conformer dataset source
>
> The conformers ${C_i}$ are supplied by the datasets, not generated by the authors. The paper closely follows the established protocols [30,51] for conformer preprocessing. For the specific datasets: Drugs-7.5K uses refined conformers from MARCEL [51], Kraken uses DFT-computed conformer ensembles [52], and CoV2 uses conformers from GEOM-Drugs [53]. We will make this clearer in the revision by explicitly stating the conformer generation protocols for each dataset.
>
> > Q3. R = SO(3) or O(3)?
>
> The paper uses R to denote the rotation component of the geometric symmetry group G (as in lines 99-100). The choice of R depends on which geometric group is selected and which GNN backbone is used:
> - When G = SE(3) (special Euclidean group), then R = SO(3) (rotations only)
> - When G = E(3) (full Euclidean group), then R = O(3) (rotations + reflections)
>
> Our model supports both options (lines 89-90) and we note that different GNN backbones correspond to different symmetry groups, e.g., ClofNet is SE(3)-equivariant (R=SO(3)). We will clearly state which geometric group and corresponding rotation group are used for each experiment.
>
> > Q4. Why $r^s$ is linear in h while $r^v$ is quadratic?
>
> This design choice is driven by symmetry preservation requirements. From Eqs. (2-3):
> - $r^s$ uses GCN aggregation: $r^\text{s}(h_i^s) = softmax(\sum_{j \in \mathcal{N}(i)} h_j^s W^s / \sqrt{d_id_j} + b)$, which is linear in $h^s$
> - $r^v$ uses inner product: $r^\text{v}(h_i^v) = softmax((h_i^v)^T h_i^v W^v)$, which is quadratic in $h^v$
>
> The quadratic form for vector features is necessary to maintain rotational invariance (Lemma 1, part 3). The operation $(h^v_i)^T h^v_i$ performs inner products over the equivariant dimension, ensuring the routing scores remain rotationally invariant while the linear form for scalar features naturally preserves invariance.
>
> > Q5. Meaning of $R^n$
>
> From the context in line 196, $R^n$ refers to the product group of $n$ independent rotational transformations, where each conformer can undergo independent geometric transformations. This is clarified in line 91: "We allow individual transformations on each conformer individually and independently, hence the overall geometric transformation $G^n$." So $R^n$ represents $n$ copies of the rotation group $R$ (either SO(3) or O(3)), not the matrix group SO($n$) or O($n$). Each conformer has its own geometric transformation capability.
>
> > Q6. Line 180 dimension reduction: $H^s$ is $N\times d$, yet $r^s$ is $N_I$
>
> $H^s$ represents node features across all conformers ($N = n\times |V|$ nodes total), while $r^s_{e,i}$ in Eq. (4) represents routing scores for the $e$-th expert and $i$-th node, with $N_I$ being the number of invariant experts. The dimensions $N$ and $N_I$ have very different meanings, and this switch occurs through the routing computation in Eqs. (2-4), where each node gets assigned scores for each of the $N_I$ invariant experts. We will clarify the dimensional flow in our revision.
>
> > Q7. Expert-routing bottleneck alleviation
>
> The routing mechanism addressed three limitations of prior uniform ensemble methods:
> 1. Uniform processing: Traditional ensemble methods treat all conformers uniformly through simple pooling, but different properties may require different conformational features (Section 3.2.2) For instance, binding affinity depends on pocket conformations while solubility depends on surface accessibility.
> 2. Symmetry preservation: The type-separated routing maintains distinct symmetry properties. A unified routing would violate physical laws by mixing R-invariant (scalar) and R-equivariant (vector) features. Separate routers ensure scalar features remain invariant while vector features preserve equivariance under geometric transformations.
> 3. Computational efficiency: Sparse activation (top-k selection) enables fine-grained expert specialization without proportional computational cost increases.
>
> > Q8. $\bar{H}$ $S_n$-invariant vs $S_n$-equivariant
>
> As in lines 223-224, $\bar{H}$ is defined as the mean-pooled representation: $\bar{H} = (1/n)\sum^n_{c=1} H^s_c$. Mean pooling across conformers breaks the equivariance to conformer permutations and instead produces an invariant representation. If conformers are permuted, the mean remains the same, making $\bar{H}$ $S_n$-invariant rather than $S_n$-equivariant.
>
> > Q9. Parameter counts and matching
>
> We ensured fair comparison by matching backbone parameters across all methods. All backbone architectures used identical configurations (hidden dimensions, number of layers, cutoff distances, RBF functions, etc.), ensuring backbone parameters remain constant with only ensemble strategies varying.
> | Backbone | Mean (M) | DeepSets (M) | Attention (M) | SPiCE (M) | Overhead (M) |
> |--|--|--|--|--|--|
> | Equiformer | 0.18 | 0.25 | 0.26 | 0.94 | 0.75 |
> | PaiNN | 0.15 | 0.22 | 0.23 | 1.16 | 1.01 |
> | ViSNet | 1.71 | 1.78 | 1.79 | 3.38 | 1.66 |
> | ClofNet | 0.52 | 0.58 | 0.60 | 2.29 | 1.77 |
>
> Note: SPiCE values represent activated parameters (8 experts, 2 activated in GMoE)
>
> Despite parameter increase, SPiCE demonstrates strong parameter efficiency through sparse activation where only a small number of experts are active during inference, significantly reducing computational cost. The consistent performance gains across all backbones and datasets justify the overhead.
>
> We acknowledge in Appendix J that “the hierarchical architecture and dependence on equivariant GNN backbones introduce additional computational overhead compared to simpler pooling methods, potentially limiting scalability in resource-constrained environments or applications requiring real-time inference”. Note that several established techniques can address these scalability concerns in future work. For example, linear attention models like Performer [1] can reduce complexity from $O(n^2)$ to $O(n)$.
>
> Additionally, expert pruning [2] could further reduce computational requirements while maintaining strong performance.
> We will include this detailed analysis in the camera-ready version.
>
> *[1] Rethinking Attention with Performers, ICLR 2021*
>
> *[2] Not All Experts are Equal: Efficient Expert Pruning and Skipping for Mixture-of-Experts Large Language Models, ACL 2023*
>
> > Q10. Performance drops for both too few and too many experts
>
> The U-shaped curve reflects the tradeoff between expert specialization and training stability.
> - Too few experts (2-4): Insufficient specialization capacity for diverse molecular patterns. Conformer ensembles contain varied geometric features (linear chains, cycles, branches, functional groups) requiring distinct computational pathways that cannot be captured with too few experts.
> - Optimal range (8-16): Provides adequate specialization while ensuring sufficient training examples per expert.
> - Too many experts (32+): Over-specialization leads to overfitting as each expert sees too few examples. The routing mechanism also becomes unstable with excessive choices, causing inconsistent expert selection.
>
> > Q11. Computational cost comparison
>
> We provide time comparison of our model with baselines as follows.
>
> *Training Time (5000 epochs using PaiNN backbone)*
> | Method | Hardware | GPU-Hours | Overhead |
> |-|-|-|-|
> | Mean | A100×2 | 10.4h | 1.0× |
> | DeepSets | A100×2 | 10.4h | 1.0× |
> | Attention | A100×2 | 10.6h | 1.02× |
> | SPiCE | A100×4 | 34.9h | 3.4× |
>
> SPiCE requires 3.4× more GPU-hours for training (configuration: 8 experts, 2 active), primarily due to the mixture-of-experts forward passes, cross-conformer attention mechanisms, and 2D topology processing. However, inference costs remain highly similar to baseline methods, showing only minimal overhead (0.03s vs 0.02s per prediction on A100). We also note that training time can be shortened when using early stopping strategies. Despite the increased training cost, the model delivers consistent performance improvements across all tasks.
>
> > Q12. $\tilde{r}_i$ dimensionality
>
> The dimensionality $\hat{r}_i \in \mathbb{R}^{N_I + N_E}$ is correct as written, since $\hat{r}_i = (\hat{r}_i^\text{s}, \hat{r}_i^\text{v})$ denotes the concatenation of routing scores for both scalar experts ($N_I$) and vector experts ($N_E$), rather than stacking. Thanks to your inquiry, we will clearly state the concatenation operation in our updated manuscript.
>
> > Q13. Loss spike near epoch 100
>
> Thank you for noting this. The loss spike corresponds to the expert upcycling transition phase (Section 3.2.2). During upcycling, additional experts are activated and routing weights are reinitialized, causing temporary optimization instability. This is expected behavior that typically resolves within 20-30 epochs as the routing mechanism adapts to the expanded expert capacity. We will clarify this in the main text.
>
> > Q14. Domain-specific shortcuts
>
> Thank you for your valuable suggestion. We will add more comprehensive background information in the appendix, including Group Theory, Equivariance and Tensor Products, to benefit readers outside the field.
>
> > Q15. Font size
>
> We will enlarge the labels and text in Figure 3 for improved readability in the revision.
>
> ---
> Thank you again for taking the time to review our paper. Please let us know if you have any further questions or suggestions.

---

> > ### Comment · Reviewer_K5sT · 2025-08-01
> >
> > Thank you for the responses. My main concerns have been resolved. I'll raise my score.
> >
> > I think the provided tables regarding num params and GPU hours are informative to readers and should be included in the Appendix. Also, the discussions of those computational costs should be noted as limitations in the main text.

---

> > > ### Author Response · Authors · 2025-08-01
> > >
> > > Thank you very much for your timely feedback and for raising your score! We will add the computational cost tables to the Appendix and will add discussion of these costs as limitations in the main text. Please let us know if you have any further questions.

---

### Official Review · Reviewer_CzgV · 2025-07-03

**Clarity:** 3
**Significance:** 3
**Originality:** 4
**Rating:** 4
**Confidence:** 4

**Summary:**

This paper introduces Symmetry-Preserving Conformer Ensemble networks (SPiCE), a new framework designed to enhance molecular representation learning by leveraging ensembles of molecular conformers rather than relying on single static structures. SPiCE designs an architecture integrating three core components: (1) shared conformer encoding with weight-tied 3D GNNs that preserve geometric equivariance, (2) a Geometric Mixture-of-Experts (GMoE) layer that processes scalar and vector features through specialized expert networks with type-aware selective routing, and (3) a hierarchical ensemble encoder that incorporates molecular-level context via cross-conformer attention. The model also provides theoretical properties of aggregations, such as R-equivariant, etc., showing the model is physically meaningful, symmetry-aware representations, and outperforms existing methods across a wide range of molecular property prediction tasks, from quantum mechanics to biology.

**Questions:**

-

**Ethical Concerns:**

["NO or VERY MINOR ethics concerns only"]

**Final Justification:**

After the rebuttal phase, I raised my score as most of my concerns  (similar to other reviewers questions) have been addressed and the explanations are very good.

**Limitations:**

-

**Paper Formatting Concerns:**

-

**Quality:**

4

**Strengths And Weaknesses:**

Strengths:

i) Method Novelty

Though the method leverages existing techniques, its overall design is well-designed, adapted to the molecular representation learning. There are also some interesting parts like differentiable top-k with a mixture of experts to selectively fuse 3D conformer to 2D representations.

Authors also present properties about the permutation invariance of the ensemble and geometric symmetries of individual conformers.

ii) Experiments:
Provided experiments are very details where SPiCe is validated on four different architectures (PaiNN, ClofNetEquiformer, and VisNet), confirming the improved performance and generalized ability of the method.
There are also experiments on model scaling on the Drug-75k dataset, at 10% and 100% training rates. A detailed of ablation studies is also provided in Figure 3.

Limitations Weaknesses:


Ambiguity in the definitions of scalar and vector features: The paper defines h_{i}^{s} ​\in R^{1\times d} as scalar features and h_{i}^{v} \in  R^{3 \times d} as vector features (line 181). However, this naming can be confusing, especially when d>1, as it may not align with the conventional understanding of scalars and vectors in the molecular context. The confusion is exacerbated by the early introduction of these terms in Figure 1. The reviewer suggests the authors consider alternative, more intuitive terminology or provide clearer justification and explanation for these definitions. Similarly, the definitions of X^s and X^v (line 151), which are inputs to the message-passing network that generates H (line 180), are not clearly explained and would benefit from further clarification.


Clarity and readability of Section 3.2.2: This section contains dense notation and is challenging to follow. The reviewer recommends augmenting this part with a relevant subfigure from Figure 1 to visually support the equations and improve reader comprehension.


Lack of discussion on limitations: The paper does not adequately discuss the limitations of SPiCE. Including such a discussion would help readers better understand the potential constraints of the method and set appropriate expectations for its applicability.


Unclear attention mechanism in Figure 1c: It is unclear why only the 3D scalar features are used as queries in the attention mechanism with 2D features. The specific role of vector features within the GMoE module should be clarified to improve the interpretability and transparency of the design choice.


Redundant discussion on equivariance and invariance: The method section repeatedly revisits the concepts of equivariance and invariance, which, while important, may distract from the understanding of other architectural components. The reviewer suggests summarizing these theoretical properties in a dedicated subsection or theorem early in the method section, allowing the rest of the space to focus on describing the novel contributions and architectural details more clearly.

---

> ### Author Rebuttal · Authors · 2025-07-31
>
> We thank you for your detailed and constructive review. Your recognition of our method’s novelty, comprehensive experimental validation, and interesting technical contributions is very encouraging. We address your specific concerns below:
>
> > Q1. Ambiguity in scalar and vector feature definitions
>
> Thank you for raising this clarity concern. Our terminology follows established geometric deep learning conventions [1] where each tensor has three distinct axes: channel ($d$), tensor-type ($l$), and tensor-component ($m$). We recognize this notation will benefit from clearer explanation. For tensor type $l$, there are $(2l+1)$ tensor components:
> * $h^s_i \in \mathbb{R}^{1 \times d}$: scalar (type-0, rotationally invariant) features with 1 tensor component and $d$ channels
> * $h^v_i \in \mathbb{R}^{3 \times d}$: vector (type-1, rotationally equivariant) features with 3 tensor components and $d$ channels
>
> The tensor-type axis ($l$) is handled by separating $h^s$ and $h^v$, which are then concatenated to form $\mathbf{X} = (\mathbf{X}^s, \mathbf{X}^v) \in \mathbb{R}^{n \times |\mathcal{V}| \times 4 \times d}$, where the 4 comes from concatenating tensor components (1+3=4). As for the definitions of $\mathbf{X}^s$ and $\mathbf{X}^v$:
> * For the first interaction block ($l=1$):
>   * $\mathbf{X}^s$ (Type-0) : Initial molecular features including embeddings of atomic numbers $z$, distance-based radial basis function $\phi_k(r)$, and topological descriptors $\mathcal{E}$
>   * $\mathbf{X}^v$ (Type-1) : Initial geometric features including embeddings of relative position vectors $\boldsymbol{\rho}$ and directional bond features
> * For subsequent interaction blocks ($l>1$), they include both the initial features above plus the processed outputs $\mathbf{X}^s$ and $\tilde{\mathbf{H}}^v$ from the previous interaction block.
>
> We will add clearer explanations of this tensor notation early in the paper and provide concrete examples to aid reader comprehension.
>
> *[1] A Hitchhiker's Guide to Geometric GNNs for 3D Atomic Systems, arXiv 2312.07511*
>
> > Q2. Section 3.2.2 clarity and readability
>
> Thank you for your feedback!  We will improve readability by adding relevant subfigures from Figure 1(d) directly into the text, providing step-by-step walkthroughs of Equations (2)-(8) with intuitive explanations, and introducing the routing mechanism in a separate subsection of expert network design.
>
> > Q3. Attention mechanism clarity (Figure 1c)
>
> Thank you for raising this question. We use only 3D type-0 features as queries because type-0 features are rotationally invariant. This ensures attention weights remain orientation-independent. Using type-1 features as queries would violate this invariance property since rotating the molecule would change attention scores, leading to inconsistent ensemble representations. In our model, type-1 features are processed separately through dedicated equivariant experts in the GMoE module to preserve their directional properties and geometric information. We will add this justification to Section 3.2.4 and revise Figure 1c with clearer annotations.
>
> > Q4. Redundant equivariance discussion
>
> Thank you for this organizational suggestion. We will restructure Section 3 by creating a single theorem early that establishes all symmetry preservation properties, then reference it throughout rather than re-explaining concepts.
>
> > Q5. Missing limitations discussion
>
> Thank you for raising this point. Due to space limitations, we included our limitations discussion in Appendix J. The key limitations include: *“While SPiCE achieves strong performance across a variety of molecular property prediction tasks, the hierarchical architecture and dependence on equivariant GNN backbones introduce additional computational overhead compared to simpler pooling methods, potentially limiting scalability in resource-constrained environments or applications requiring real-time inference. Furthermore, although SPiCE facilitates cross-conformer interactions through attention-based integration, the framework does not explicitly incorporate thermodynamic priors or statistical mechanical principles that govern conformational ensembles.”*
>
> We will move limitations to the main paper in the final version.
>
> ---
> Thank you again for taking the time to review our manuscript. Please let us know if you have any additional questions.

---

> > ### Comment · Reviewer_CzgV · 2025-08-05
> >
> > I agree with the author's clarifications.

---

> > > ### Author Response · Authors · 2025-08-05
> > >
> > > Thank you very much for your constructive feedback. We are pleased that our response has addressed your concerns. We will revise our paper and add those discussions.

---

> ### Comment · Area_Chair_jLzF · 2025-08-05
> **Please respond to authors' rebuttal before Aug. 6 (AOE)**
>
> Dear Reviewer CzgV,
>
> This is a reminder that the author-reviewer discussion period is ending soon on Aug. 6 (AOE), and you have not yet responded to authors' rebuttal. Please read authors' rebuttal as soon as possible, and engage in any necessary discussions, and consider if you would like to update your review and score. Please at least submit the Mandatory Acknowledgement as a sign that you have completed this task.
>
> Thank you for your service in the review process.
>
> AC

---

### Official Review · Reviewer_LtiX · 2025-07-06

**Clarity:** 3
**Significance:** 4
**Originality:** 4
**Rating:** 5
**Confidence:** 4

**Summary:**

The paper introduces a new architecture for conformer-based molecular property prediction. This architecture involves a Mixture of Experts for analyzing the latent features of the molecule. Care is taken to differentiate between the vector and scalar components, and it is shown that the architecture preserves the equivariance/invariance of the underlying experts (GNNs). The equivariant experts are 3D GNNs, and the invariant experts are 2D topological GNNs. Attention is used to combine the information of both types of experts.

**Questions:**

Is it possible that the size of the datasets used in evaluation impacted the performance of the model? At the largest, 75k molecules/560k conformers is only a moderately sized dataset, and most of the comparisons were done with a model trained on a fraction of those data.

Possible confusion may arise with the similarly named SPICE dataset (https://www.nature.com/articles/s41597-022-01882-6).

**Ethical Concerns:**

["NO or VERY MINOR ethics concerns only"]

**Final Justification:**

As my initial rating was a 5, the discussion with the authors was mainly to confirm my original evaluation. They did address a couple of my concerns. I decided to keep my rating.

**Limitations:**

yes

**Paper Formatting Concerns:**

None.

**Quality:**

3

**Strengths And Weaknesses:**

Quality: The submission takes care to explain the technical details of the architecture and all of the mathematical proofs that the architecture is invariant/equivariant in the appropriate places. The work did not really cover weaknesses of the introduced architecture. In particular, the datasets evaluated seem relatively small.

Clarity: The writing was generally clear, and the level of detail in section 3 was good. Sections 4 and 5 felt brief in comparison to how dense section 3 was.

Significance: The work improves the state of the art on some molecular property prediction tasks, and the details are explained well enough that this could be extended in the future. The method of this work is more general than many previous works, and so the mathematical foundation developed in this work can likely be used in later works to extend and improve the results described here.

Originality: The comparisons to other similar conformer-based and non-conformer-based models are helpful in providing context. It improves on and generalizes the previous works.

---

> ### Author Rebuttal · Authors · 2025-07-31
>
> We sincerely thank you for your positive assessment and recognition of our work’s technical rigor, clarity, generalizability to future research, and improvements to molecular property prediction.
>
> > Q1. Dataset size impact on model performance
>
> Thank you for raising this important consideration about data scale. We agree that evaluation on large-scale datasets will strengthen our claims. However, we would like to note that many molecular property prediction tasks are of similar or smaller scales due to experimental costs, making our evaluation practically meaningful.
>
> Additionally, our evaluation demonstrates consistent improvements across diverse dataset sizes (1.5K to 75K molecules), and we studied the scaling of training data to model performance. As shown in Section 4.3 and Figure 2(a), SPiCE exhibits approximately linear scaling from 7.5K to 75K molecules, reducing MAE from 0.4929 to 0.4386. This suggests that SPiCE maintains stable learning characteristics regardless of data scale.
>
> > Q2. SPICE naming confusion
>
> Thank you for flagging this potential confusion with the SPICE dataset from Eastman et al. (2023). We will rename our method to **SPECE** to avoid ambiguity.

---

> > ### Comment · Reviewer_LtiX · 2025-08-06
> > **Thank you for your response**
> >
> > As my original rating was 5, I will keep that.

---

> > > ### Author Response · Authors · 2025-08-06
> > >
> > > Thank you very much for your feedback and your positive assessment! Please let us know if you have additional questions.

---

> ### Comment · Area_Chair_jLzF · 2025-08-05
> **Please respond to authors' rebuttal before Aug. 6 (AOE)**
>
> Dear Reviewer LtiX,
>
> This is a reminder that the author-reviewer discussion period is ending soon on Aug. 6 (AOE), and you have not yet responded to authors' rebuttal. Please read authors' rebuttal as soon as possible, and engage in any necessary discussions, and consider if you would like to update your review and score. Please at least submit the Mandatory Acknowledgement as a sign that you have completed this task.
>
> Thank you for your service in the review process.
>
> AC

---

### Decision · Program_Chairs · 2025-09-17

**Decision:**

Accept (poster)

**Comment:**

The submission proposes augmenting the model input with 3D conformation ensemble samples of the input molecule for predicting molecule-level (i.e., not depending on a specific conformation of the molecule but determined by the thermodynamic distribution of the conformations) properties. The authors designed some model components (e.g., geometric mixture-of-experts, invariant and equivariant experts, inter-conformation interaction) that seem reasonable, and geometric and permutation symmetries are tracked along with the model development. Empirical results demonstrate better performance than models that do not use conformation or use a single conformation.

The authors and I appreciate the idea of leveraging detailed conformation ensemble information for predicting molecule-level properties, and the design seems reasonable and effective. Therefore, I'd recommend an acceptance considering the (one of, if not exactly) the first work that embarks this route. Nevertheless, I hope the authors could improve the paper according to reviewers' feedback, particularly the acronym, notation, presentation, and performance-cost trade-off comparison results. Moreover, I also sensed a few points that I suppose are worth discussing:
* Should the learned ensemble/molecule representation and prediction model be affected by the distribution of the provided samples? I suppose it should be (different distributions, e.g., under different temperatures, should affect ensemble/molecule-level properties), but the authors are expected to discuss how to ensure this. Particularly, in aggregating conformer-level representations, it seems that the conformers are equally weighted, but the samples are usually not following the Boltzmann/thermodynamic equilibrium distribution (e.g. in the GEOM case).
* The authors are expected to expand the definitions of the predicted properties and explain why they are ensemble/molecule-level properties.